palaeontology/geochemistry/materials science

conodont, ontogeny, microstructure, diffraction, mosaicity, texturing

**Authors for correspondence:**
Bradley D. Cramer
e-mail: bradley-cramer@uiowa.edu
Tori Z. Forbes
e-mail: tori-forbes@uiowa.edu

# Ontogenetic variability in crystallography and mosaicity of conodont apatite: implications for microstructure, palaeothermometry and geochemistry

Mohammad Shohel[1], Neo E. B. McAdams[3],
Bradley D. Cramer[2] and Tori Z. Forbes[1]

[1]Department of Chemistry, and [2]Department of Earth and Environmental Sciences, University of Iowa, Iowa City, IA 52242, USA
[3]Department of Geosciences, Texas Tech University, Lubbock, TX 79409, USA

(iD) MS, 0000-0003-2736-2389; TZF, 0000-0002-5234-8127

X-ray diffraction data from Silurian conodonts belonging to various developmental stages of the species *Dapsilodus obliquicostatus* demonstrate changes in crystallography and degree of nanocrystallite ordering (mosaicity) in both lamellar crown tissue and white matter. The exclusive use of a single species in this study, combined with systematic testing of each element type at multiple locations, provided insight into microstructural and crystallographic differentiation between element type ($S_a$, $S_{b-c}$, $M$) as well as between juveniles and adults. A relative increase in the unit cell dimensions $a/c$ ratio of nanocrystallites during growth was apparent in areas demonstrating single-crystal behaviour, but no such relationship was seen in dominantly polycrystalline areas. Systematic variations in mosaicity were identified, with mosaicity (as a proxy for disorder) increasing during growth, as well as along elements from tip to base. These results provide potential insight into the integrity of conodont apatite as a recorder of palaeoseawater chemistry, as well as demonstrate the need to consider the influence of ontogeny and element type on the use of conodonts in palaeothermometry and geochemical investigations.

# 1. Introduction

Bioapatites from conodont microfossils are routinely used to develop basin-scale thermal histories [1], for palaeothermometry of the ocean–atmosphere system [2–5], and are increasingly used as a proxy for ancient ocean chemistry [6–14]. They are particularly useful, given their biostratigraphic significance and nearly ubiquitous presence in marine strata, and numerous studies have illustrated that conodont apatite is more resistant to diagenetic alteration than is brachiopod calcite [4,5,15]. Even seemingly pristine brachiopods screened using established trace element discrimination methods can be geochemically altered [4,15,16], which has promoted the use of conodont bioapatite for reconstruction of palaeoseawater chemistry. Despite extensive use, it has been found that choice of proper conodont tissue could be critical in palaeo reconstruction as extent of post-mortem chemical alteration depends on tissue type, thus may give contradictory chemical signature from different spots of same sample [14,17]. However, some uncertainties remain regarding conodont bioapatite formation processes, potential vital effects on elemental and isotopic compositions, the role of microstructure on chemical retention, and ontogenetic variations in crystal chemistry.

Early studies identified two distinct types of crown tissue in conodonts and a less commonly preserved distinct tissue type of the basal body [18,19]. The translucent lamellar crown tissue, and typically opaque, white matter comprise the crown tissue of euconodonts [20–22]. All tissue types are composed of nanocrystals of apatite ranging from 10s to approximately 100 nm wide by 0.1–10 µm long that are embedded in an organic matrix [21–25]. The ratio of nanocrystal to organic matrix varies between the three tissue types with white matter typically being the most crystal rich (or behaving as a single crystal) and containing the least amount of organic matrix. Whereas the significance of phylogeny and ontogeny in microstructure has been well documented [22,26,27], many studies of conodont crystallography combine crystallographic data from different species, different element types, different intervals of Earth history and different stages of ontogenetic development [24,28–30]. The net result is aggregated data that do not display significant trends and a conclusion that conodont bioapatite crystallography is essentially identical across the clade. Here, we focus on a single species of coniform-bearing conodont, *Dapsilodus obliquicostatus* [31], by using two-dimensional micro X-ray diffraction techniques that allowed us to examine discrete portions of each conodont element. We examined 14 specimens and analysed 63 total spots through micro X-ray diffraction from multiple positions along each specimen. By systematically evaluating the differences between element type ($S_a$, $S_{b-c}$, M), as well as the differences between growth stages of each element type, we can address the role of ontogeny in both crystallography and the structural ordering (mosaicity) of nanocrystallites within conodont bioapatite.

# 2. Material and methods

Conodont samples used in this study came from the Schlamer #1 drillcore, Alexander County, southwestern Illinois, USA [32]. All specimens analysed were recovered from the St Clair Formation and are well preserved and thermally unaltered with a conodont colour alteration index (CAI) of 1 indicating a burial temperature no higher than 80°C [1]. The low CAI of these specimens demonstrates a very low likelihood of pyrolysis of the original organic matter in the bioapatite due to metamorphism. The St Clair Formation records the Wenlock Epoch of the Silurian Period and spans an interval from approximately 433 to 426.75 Ma [32,33]. Carbonate and carbonaceous shale samples from the core ranged in size from 7.5 to 15 cm in stratigraphic thickness and were digested in the University of Iowa Micropaleontology Laboratory using the standard double-buffered formic acid technique [34]. Insoluble residues were further processed by heavy liquid separation using lithium metatungstate (LMT) at a density of 2.83–2.84 kg l$^{-1}$, and the remaining heavy fraction was picked under binocular microscope for conodonts. Specimens of *D. obliquicostatus* were selected to represent a range of ontogenetic development from juvenile to gerontic (figures 1 and 2), and ontogenetic development was determined by appearance (size, robustness, apparent wear, transparency). The overwhelming abundance of *D. obliquicostatus* in the studied core provided more than 1000 elements of this species clearly demonstrating growth series (figure 1) and was critical to this study. This abundance provided hundreds of specimens of each element type, from a range of ontogenetic development, from a single species, from a single locality and from a single window of Earth history, which allowed us to control for many of the parameters that are often under-evaluated in other studies. The elements chosen for XRD analysis were grouped into two categories, juvenile or older

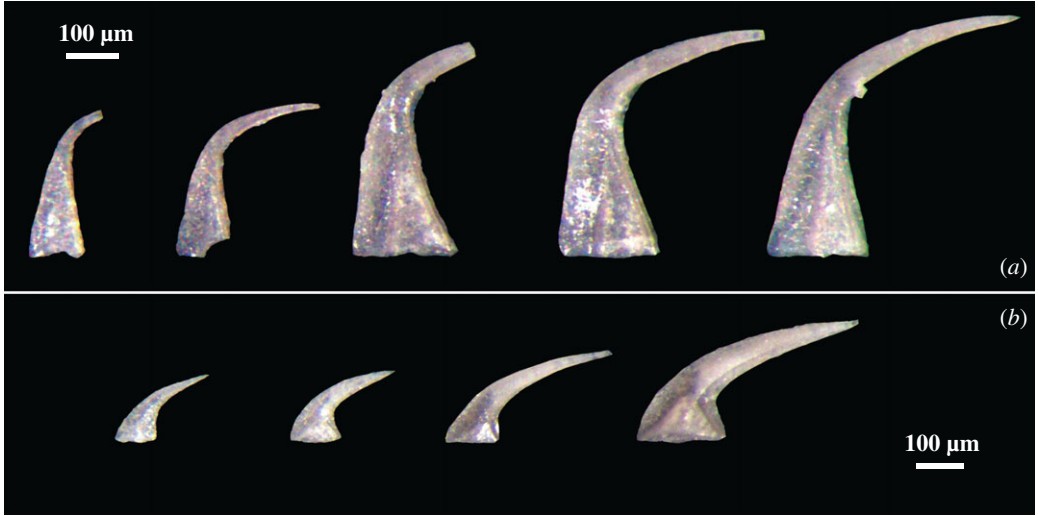

**Figure 1.** Representative growth series S and M elements of *D. obliquicostatus* presented synoptically such that each panel has a constant scale for each specimen. (*a*) $S_a$ elements and (*b*) M elements. Scale bar in each panel is 100 µm.

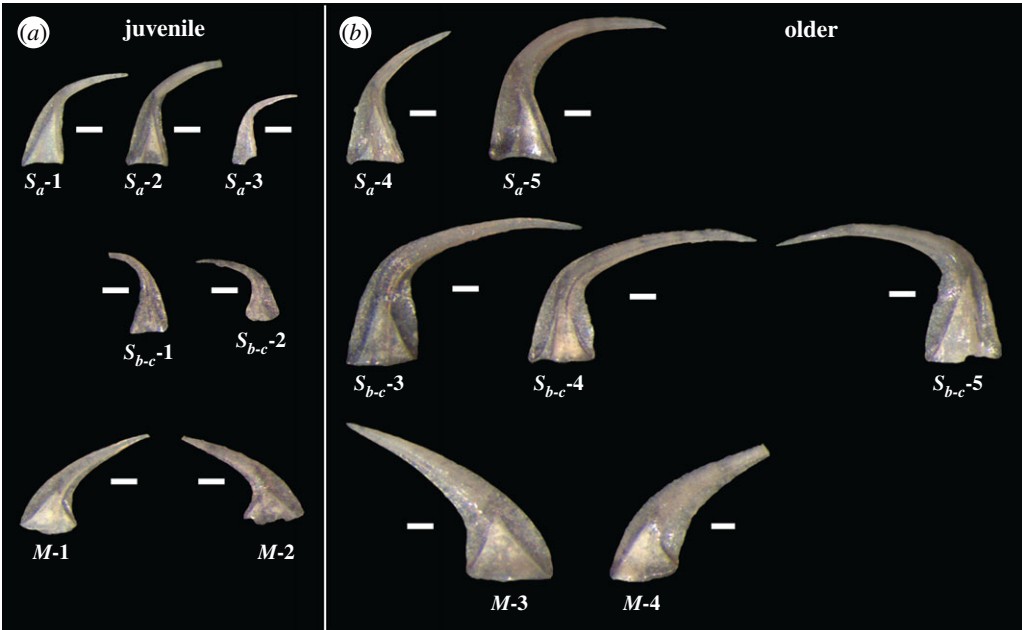

**Figure 2.** Conodont elements of *D. obliquicostatus* used in the present study. Each scale bar corresponds to 100 µm and the figure is presented synoptically to provide a clear demonstration of the size of each specimen compared with the others. (*a*) Juvenile elements and (*b*) older elements.

(figure 2). The differentiation into each category was done primarily on the overall size of the element, with secondary affirmation provided by the characteristics of robustness, wear and transparency, as well as morphological traits such as degree of development of lateral costae, the extension of the posterior keel or expansion of the basal cavity.

The apparatus of *D. obliquicostatus* [35] includes an acodontiform *M* element, a distacodontiform $S_a$ element modified with lateral costae that occasionally extend behind the posterior keel, and distacodontiform $S_b$ and $S_c$ elements that exhibit slight to strong twisting along the cusp [36]. The degree of twisting is the basis for differentiation between $S_b$ and $S_c$ elements, but given the lack of a precise definition combined with the change exhibited over time, most authors tabulate a combined $S_{b-c}$ element type [36] and we have followed this procedure herein. The entire apparatus has not been found *in situ*, but approximations of elements in the apparatus based upon ratio counts indicate an apparatus with a ratio of $1S_a : 10S_{b-c} : 5M$ [32,35–37]. None of the specimens included in this study included a preserved basal body.

X-ray diffraction experiments were carried out in a Bruker D8 Quest single-crystal X-ray diffractometer equipped with a CMOS area detector that allows diffraction pattern analysis of both single crystalline and polycrystalline material. Irradiation of samples occurred through a microfocus Mo K$\alpha$ ($\lambda = 0.7107$ Å) X-ray source with a beam diameter of 120 µm, which is suitable to analyse multiple positions along each specimen. Depending on the size of the sample, three to six zones were analysed along the length of each element. In each experiment, conodont elements were coated with mineral oil, then attached on top of MiTeGen Dual-Thickness MicroLoop in a vertical position. The sample was placed on the three-circle goniometer and the position of the X-ray beam was chosen using the microscope camera.

X-ray diffraction data associated with each position were collected in transmission mode by varying the $\Phi$ angle while keeping the other angles fixed ($2\theta = 0°$, $\omega = 0°$ and $\chi = 54.74°$). An 8–11 s exposure time was needed for analysing lamellar crown tissue and white matter and a 20–30 s exposure time was needed for the basal cavity. All experiments were carried out at 100 K to reduce thermal motion of atoms. For single crystalline zones, unit cell parameters were determined by identifying the well-resolved reflections within the two-dimensional frames, followed by indexing and refinement of the $a$ and $c$ parameters using Bruker APEX3 software. Mosaicity values represent the full-width half-max of the reflections and are calculated automatically by the APEX3 software during the unit cell refinement process. For polycrystalline zones, the Debye rings were integrated using APEX3 software to create the resulting powder X-ray diffraction (PXRD) pattern. Unit cell parameters were obtained by indexing the PXRD pattern by the least square cell parameter program [38]. XRD peaks that were well separated and did not overlap with other peaks were selected for indexing. Alignment of the diffractometer was calibrated using the standard YLID crystal provided by Bruker AXS. Detector distance, swing angle and beam centre were calibrated by comparing the $2\theta$ angles and detector position of a polycrystalline biomimetic fluorapatite standard produced in the University of Iowa Department of Chemistry [39,40]. Preferred crystallographic orientation was analysed by Psi scan (also known as $\beta$ angle) using XRD2DScan software [41]. A diagram of the equipment and design of the X-ray diffraction experiment is provided in the electronic supplementary material, figure S1.

Quantile–quantile (qq) plots of single crystalline zones are provided in electronic supplementary material, figure S2 as a test of normal distribution of unit cell parameters. Given that neither the measurements on $a$ nor $c$ are truly 'controlled' and thus each independent variables, model II linear regressions (ordinary least squares (OLS), major axis (MA), standard major axis (SMA) and ranged major axis (RMA)) were carried out to determine the association of the unit cell parameters. A model II OLS regression is the bifurcation of the '$x$ on $y$' and '$y$ on $x$' regressions. Parametric $p$-values for OLS and permutational probability tests (for OLS, MA and RMA) were calculated ($n = 10\,000$) to determine whether the correlations were statistically significant. Regression analysis was performed in 'R' using the 'lmodel2' package [42]. Following the decision tree in Legendre & Legendre [43] and Legendre [42], the random variation (i.e. error variance) on both variables ($a$ and $c$) are approximately equal, the data have an approximately bivariate normal distribution (electronic supplementary material, figure S2), and both variables are expressed in the same physical units; therefore, MA regression should be used and is shown below. All regression types are shown in electronic supplementary material, figures S3 and S4 and tables S6 and S7 (see Discussion below).

# 3. Results

## 3.1. Two-dimensional X-ray diffraction and unit cell parameters

Crystalline materials interact with X-ray radiation to give a characteristic diffraction pattern due to coherent scatting of these electromagnetic waves. Materials typically lie within a spectrum of crystallinity that ranges from the ideal single crystalline to polycrystalline and finally amorphous, which are defined by the relative length of the coherent scattering domain, the relative disorder of those domains and their overall orientation [44,45]. Materials composed of highly ordered atomic lattices (e.g. single crystals) display diffraction individual reflections (observed as spots on the two-dimensional area detectors) that are indicative of the large domains that diffract with coherent scattering [46]. Polycrystalline materials are composed of small (5–10 µm) crystallites that are arranged in random orientations throughout the sample [44]. Coherent scattering of the X-rays in a polycrystalline sample results in a series of diffraction rings (Debye rings). Amorphous materials may have large particle sizes, but the coherent scattering domains are small (less than 5 nm) and result in broad, low-intensity features in the

two-dimensional images [44–46]. Disorder in coherent domains of single crystalline structures will change the angle of diffracted X-rays corresponding to related crystallographic planes in varying degrees; thus will change the shape and width of diffraction features [44,47]. In the case of polycrystalline materials, if the small single crystallites are not randomly oriented then some of the crystallographic planes will interact with incoming X-rays more often, thus diffracted X-rays from those planes will create more intense regions in the diffraction pattern [48]. Variations in the intensity of the Debye rings in diffraction pattern are a phenomenon known as texturing [48]. As X-rays interact differently depending on crystalline structure, we determined unit cell parameters of conodont zones using separate protocols suitable for either single crystalline or polycrystalline material.

Highly crystalline and polycrystalline regions within the conodont samples were analysed in different ways due to the differences in the scattering. For highly crystalline zones that contained individual reflections, the reciprocal lattice was converted into direct lattice using a Fourier transform and then unit cell parameters could be calculated based upon the positions of those reflections. Within polycrystalline zones, Debye rings in the diffraction pattern were integrated, after which the $d$-spacing for each diffraction peak was calculated and used to determine unit cell parameters. For some zones, unit cell parameters and other crystalline parameters (indicated with NR in electronic supplementary material, tables S1–S3) could not be calculated due to (i) overlap of the Debye ring and single-crystal reflections, (ii) significant texturing of the Debye rings, which left determination of unit cell parameters by either of our protocols unsuitable, or (iii) insufficient number of diffraction spots.

The two-dimensional diffraction patterns indicate variability of crystalline structure along the length of $D.$ $obliquicostatus$ elements (figure 3). For juvenile $S_a$ and $S_{b-c}$ elements, we observed single and separated reflections in the diffraction pattern from both lamellar crown tissue and white matter, indicating an ordered single crystalline apatite structure in both tissue types. By contrast, older $S_a$ and $S_{b-c}$ elements displayed elongation and texturing of the individual reflections within both lamellar crown tissue and white matter areas, indicating some level of disorder in coherent domains of diffraction. All $M$ elements possess single and separated reflections within both lamellar crown tissue and white matter regions despite their ontogeny, indicating a highly ordered crystalline structure and a clear differentiation from S elements in this species. The basal cavity of all conodont elements in our study are polycrystalline as evident by the presence of Debye rings in two-dimensional diffraction patterns; however, we left the basal cavity intact for all samples (i.e. there was the open space of the cavity itself). Interference resulting from X-rays passing through the outer white matter portion of the basal cavity, into the open space of the cavity and into the opposite outer lamellar crown tissue portion of the basal cavity may pose a question about the Debye ring formation. Therefore, we ran an additional experiment on a subsample of bioapatite taken from a single side of the basal cavity and also observed Debye ring formation, proving inherent polycrystallinity in that zone.

Calculated unit cell parameters for single crystalline and polycrystalline regions of conodont elements are shown in figure 4. The average unit cell parameter $a$ for polycrystalline zones was within error of single crystalline zones with values $9.385 \pm 0.017$ and $9.371 \pm 0.018$ Å, respectively. The values were lower than the $a$ value of Durango apatite and volcanic apatite minerals analysed in our study and other literature values (electronic supplementary material, table S4). A similar relationship was observed for $c$ with averages of $6.870 \pm 0.021$ Å for polycrystalline zones and $6.892 \pm 0.015$ Å for single crystalline regions. The distribution of single crystalline zone data from juvenile versus older specimens (figure 4$a$) illustrates what appears to be separation between the two groups. Model II OLS linear regressions (electronic supplementary material, table S6 and figures S3 and S4) of each population yielded positive correlation coefficients of $r = 0.7314171$ (two-tailed parametric $p$-value = 0.001943149) for juvenile specimens and $r = 0.8567035$ (two-tailed parametric $p$-value = 0.00000142126) for older specimens. The 'lmodel2' package provides permutational probability tests for the MA regression as a one-tailed permutation test $p$-value in the direction of the sign of the slope, and we chose $n = 10\,000$ for the number of permutations. For single crystalline zones, the one-tailed $p$-value for the MA regression was 0.0034 for juvenile specimens and 0.0001 for older specimens.

Whereas the limited data provided by this study suggest that there may be systematic change in the unit cell dimensions $a/c$ ratio with respect to ontogeny in single crystalline zones due to a relative increase in the $a$ length compared with the $c$ in older specimens, there are several important issues to discuss. First, the artificial separation of a continuous series (i.e. growth) into two groups here (juvenile and older) may be showing a greater degree of separation between these two groups than if we had a more continuous series within a much larger set of data. Secondly, whereas the significance tests for each group's regressions demonstrate a fairly high degree of correlation, that is not the same as demonstrating statistical significance of separation between the two groups (juveniles

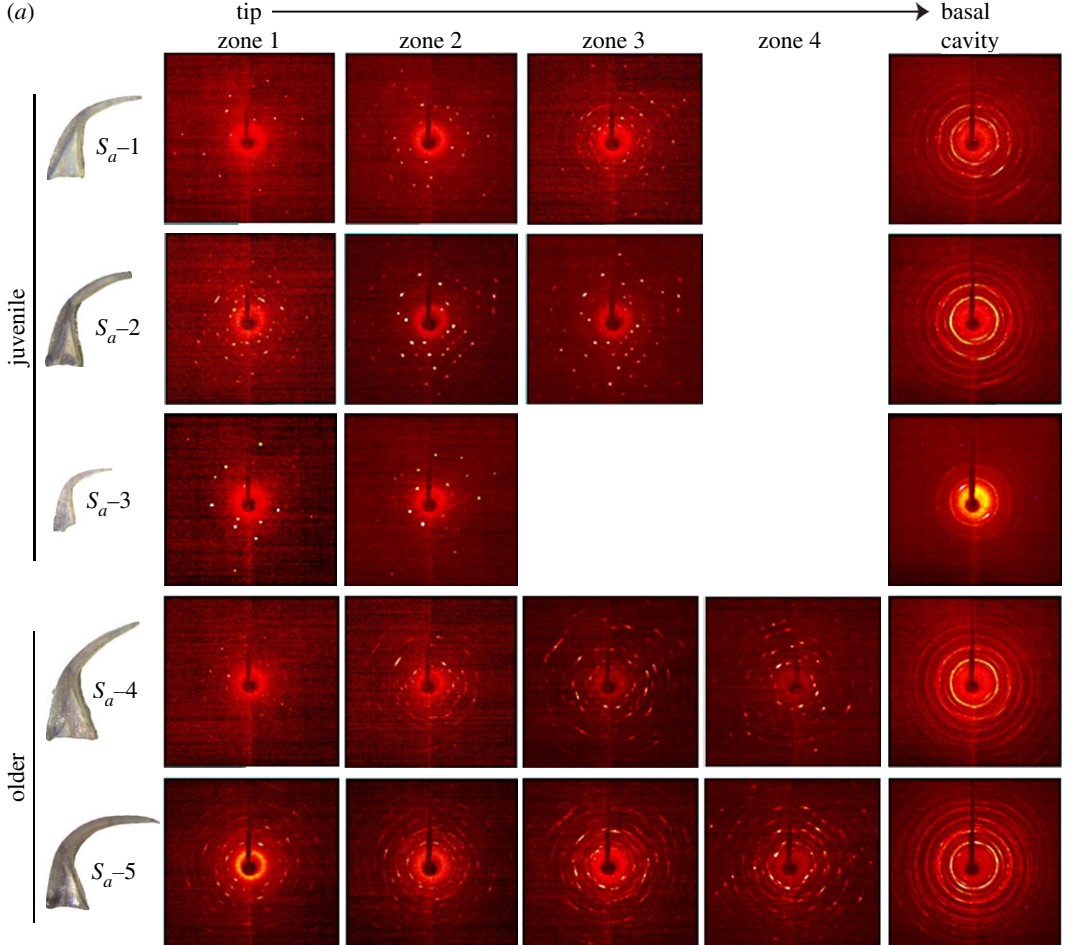

**Figure 3.** Representative two-dimensional X-ray diffraction patterns showing variation of crystalline structure at different zones of juvenile and older elements along the length (a) $S_a$, (b) $S_{b-c}$ and (c) M. The basal cavity was polycrystalline as seen from Debye rings in the diffraction pattern of the last zone. Other zones in white matter were single crystalline with separate spots in the diffraction pattern. For older $S_a$ and $S_{b-c}$ elements, elongation of the reflections (i.e. smearing of the spots) was observed in single crystalline zones. Note the greater prevalence of single crystalline (i.e. discrete spots) diffraction patterns in all M elements in (c) compared with (a,b).

and older). The limited size of the dataset cannot conclusively demonstrate statistical significance of the difference between these two populations, and the 95% confidence intervals on the two MA regressions show considerable overlap (electronic supplementary material, figure S5) even within the range of our data. Whereas the data may suggest that the unit cell parameters may be changing during ontogeny, and that this is an intriguing result that certainly warrants future study, the limited sample size of the data presented are a pilot study at best and should be considered preliminary regarding this relationship.

## 3.2. Mosaicity and texturing

Mosaicity describes the level of crystalline ordering observed within the material and can be used as a metric describing that crystallinity. Perfect single crystals have no defects within the lattice and unit cells can be described as an infinite three-dimensional tiling. However, most crystals are not perfect and contain defects that then cause subtle misalignment of the stacked unit cells (domain of coherent X-ray scattering). Therefore, the mosaicity of the material describes the degree of crystalline imperfections and allows us to describe the conodont sample as composed of mosaic blocks with different levels of alignment [49]. The value of mosaicity we report is dependent on the spread of reflections observed within the X-ray diffraction and described by evaluating the full-width half-max of the reflection when evaluating the intensity versus rotation angle plot [50,51]. The average mosaicity of conodont single crystalline zones was $1.26 \pm 07°$, which was higher than natural apatite minerals ($0.955 \pm 0.005°$), suggesting crystalline imperfections due to disorder (figure 5). Moreover, the

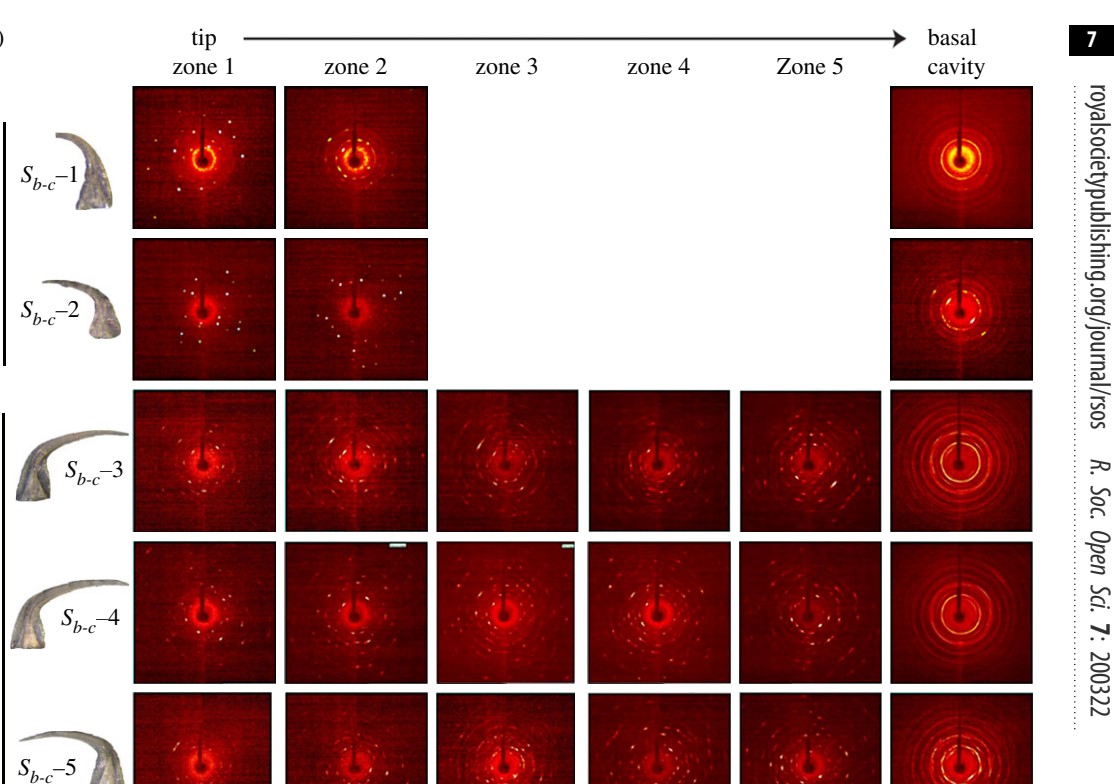

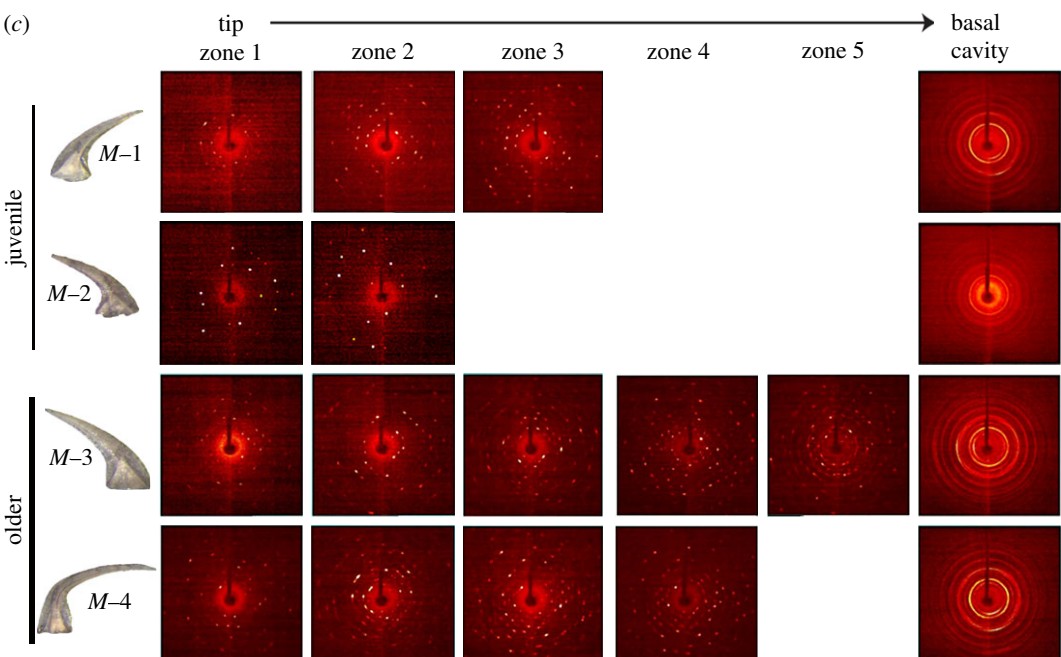

**Figure 3.** (Continued.)

mosaicity of juvenile elements ($1.21 \pm 0.07°$) was lower than older elements observed from their average values for all zones ($1.30 \pm 0.05°$). The disorder in crystalline structure was also found to increase going from tip to basal cavity as seen from the trend in mosaicity values (figure 5a).

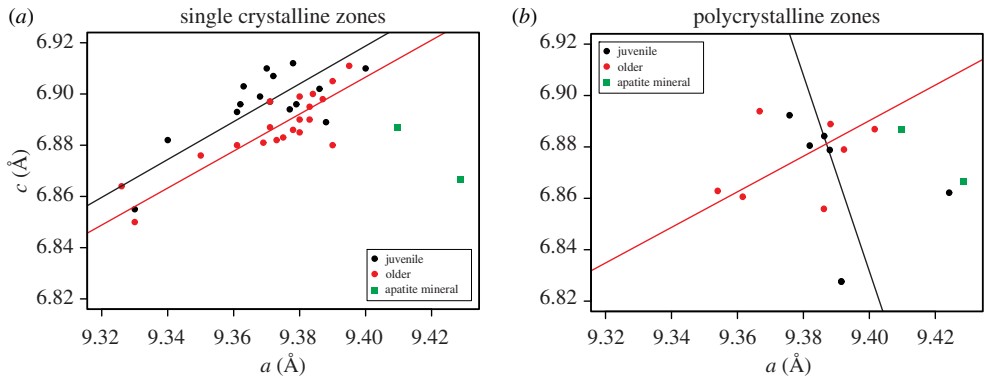

**Figure 4.** Binary plots showing unit cell parameters of conodont elements refined from (*a*) single crystalline zones and (*b*) polycrystalline zones. Unit cell parameters of the Durango apatite and a volcanic apatite measured by this study are also shown for comparison. Model II major-axis regressions are shown in each panel; however, they are meaningless in (*b*) due to the extremely small sample size. Four model II regressions were run on each grouping of data, including OLS, MA, SMA and RMA. See electronic supplementary material for regression analysis, plots and 'R' program calls using the 'lmodel2' package.

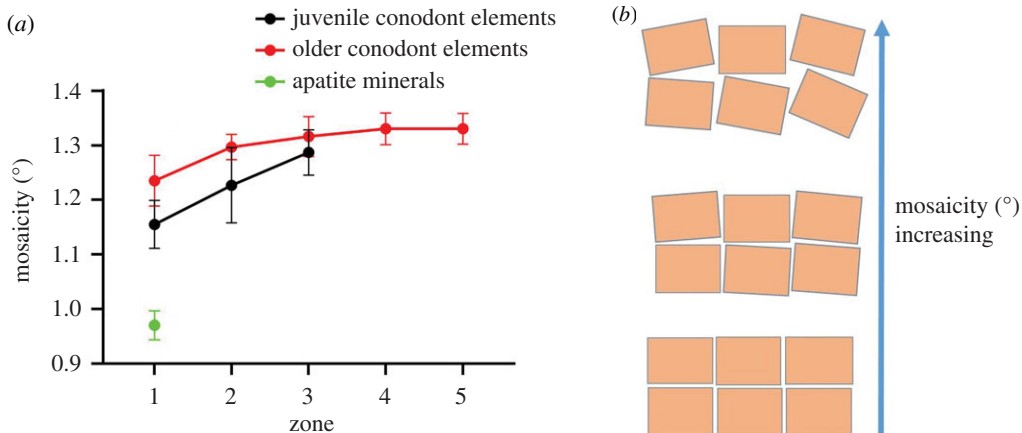

**Figure 5.** (*a*) Mosaicity values at single crystalline zones of elements show systematic increase when going from tip to basal cavity. However, juvenile elements have lower mosaicity than older ones at similar zones. (*b*) An illustration of crystalline domains at different degrees of relative mosaicity. The orientation of the crystalline domains is exaggerated in this image to illustrate the idea of mosaicity.

An ideal polycrystalline material is made of small single crystallites arranged in a manner that all the crystallographic planes are randomly oriented. However, crystallites without random orientation will make some crystallographic planes more likely to interact with incoming X-rays during analysis, which will result in texturing in the diffraction pattern. All of the conodonts that we analysed showed similar texturing in polycrystalline zones as evident from intense brighter spots within some Debye rings (figure 6). Psi scans of the (002) and (310) planes were used to analyse texturing by integrating intensity profile along the Debye ring sector as a function of $\Psi$ angle (also known as $\beta$ angle). The Psi scans resulted in two peaks separated by approximately 100° for both the (002) and (310) planes (figure 6), with more texturing observed for (002) than (310). The position of texturing in the diffraction pattern relative to the $\Psi$ angle was also different for the two planes. Psi scans at different phi angles showed variability in position and intensity of texturing around each polycrystalline region. Preferred crystallographic orientation at the 002 plane was also observed previously in tooth enamel, bone and conodonts [23,52,53].

## 4. Discussion

Previous studies on coniform elements suggested that white matter is made of large compact crystals arranged along the *c*-axes perpendicular or sub-perpendicular to the long axis of the element, such that the entire crown can be treated as single homogeneous prism of crystallites [21,23]. The ordering

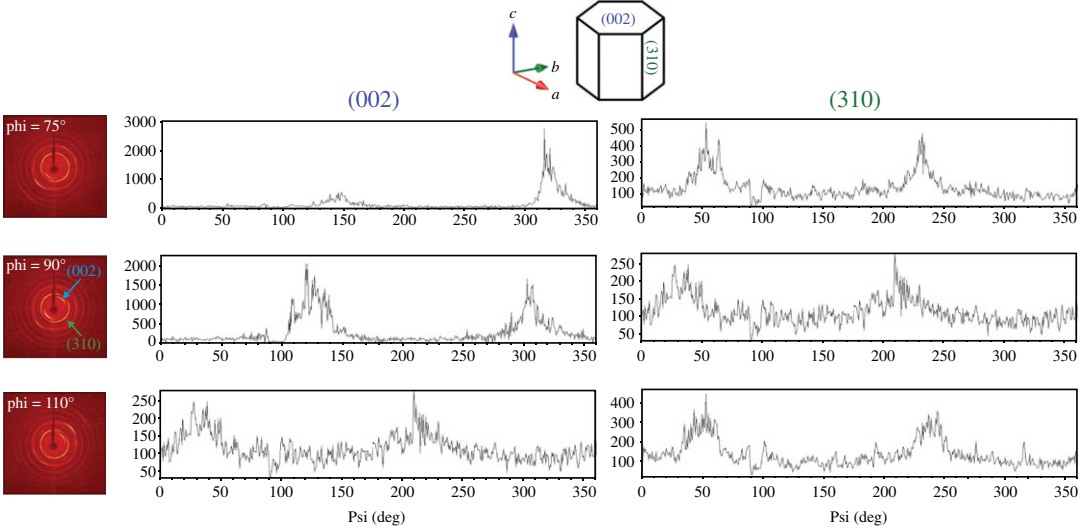

**Figure 6.** Texturing in polycrystalline regions of conodont element ($S_{b-c}$—4) at different phi angle seen from two-dimensional diffraction pattern and Psi scans shows preferred crystallographic orientation for (002) and (310) planes.

of large crystals in a single crystallographic direction explains the separate diffraction spots in the XRD pattern from single-crystal zones of elements studied herein. We also observed both single reflections and Debye rings in lamellar crown tissue of some samples, indicating a transition from aligned domains to a less ordered structure. Whereas previous studies identified this combination of single reflections and Debye rings by using a 300 µm X-ray beam to study the whole conodont element at once [28], the smaller beam size used in our study allowed us to identify the difference between tissue types and illuminate this variability in crystallographic structural detail.

The XRD patterns in figure 3 illustrate that the M elements of *D. obliquicostatus* are substantially less polycrystalline than are any of the S elements, regardless of the stage of ontogenetic development. The M elements exhibit almost exclusively single-crystal domains in the XRD pattern, even in lamellar crown tissue. This differs from the typical single crystal/polycrystalline-white matter/lamellar crown tissue relationship of S elements and suggests a unique growth behaviour of M elements within the apparatus of this species. Such microstructural variations have been identified between different species of different genera [21,22] and are probably a functional adaptation to structural/rheological requirements of each element [22]. Our data illustrate differences in microstructural growth patterning and crystallographic behaviour between different elements of a single species, which adds further support to the role of functional adaptation in crystallographic microstructure of conodont bioapatite [22,54].

Previous studies also used X-ray diffraction techniques to extract important information on mineralogy and diagenesis of conodont bioapatite [24,28–30,55]. Notable work by Zhang *et al.* [29] investigated 19 samples from coniform conodonts using high-resolution X-ray diffraction and found that the crystallinity varied from poorly ordered amorphous structure to more crystalline regions, but did not observe systematic trends related to chemical composition. In addition, Medici *et al.* [28] evaluated unit cell parameters of conodont bioapatite across geologic age, taxononomy, geographical location and CAI values. The aggregation of their data revealed that the bioapatite unit cell varied over the samples. The unit cell parameter *a* of euconodonts from their study was found to be higher than paraconodonts and non-conodont phosphatic fauna. We observed a similar trend where the overall unit cell parameter *a* for our study was higher than paraconodonts but generally within the range of post-Cambrian euconodonts [24,28–30,55]. Interestingly, however, a few outliers from M and $S_a$ elements were found to have *a* parameters that were significantly lower than previously measured euconodonts.

The unit cell parameters of biogenic apatite are governed by substitution of anionic complexes into the phosphate site and metal cations into calcium sites [Ca(I) and Ca(II)] within the crystal lattice [55–58]. The unit cell parameter *a* of synthesized pure fluorapatite has been found to be significantly lower than natural bioapatite due to substitution of hydroxyl and carbonate ions [28,57]. Carbonate substitution at the phosphate site is responsible for expansion of *c* and contraction of *a* [59]. In addition, thermal expansion behaviour of fluoro- and chloroapatite minerals can cause a systematic

increase in both $a$ and $c$ parameters with a sensitivity to chemical composition of the material [60]. There are also possible chemical substitutions on the cation sites that could influence the unit cell parameters. Due to high flexibility of the calcium site in apatite structure, conodont elements can accumulate cations of different sizes and charges during their life cycle and post-mortem diagenesis [10,12,14,29,60]. The effects on lattice parameters due to substitution of cations in apatite are complex and depend on various factors including size, charge and concentration of cations [61]. Ionic radius governs the substitution of cations in both the Ca(I) and Ca(II) sites within the apatite lattice, where cations with a radius larger than $Ca^{2+}$ tend to occupy site (II) because the Ca(II) site has a larger volume than the Ca(I) site [62,63]. In the case of substitution of cations having charge other than 2+, a charge neutrality needs to be created through vacancy formation or co-substitution of other ions at the Ca site [63]. Moreover, the type of anionic substitution also influences the cationic distribution between the two sites Ca(I) and Ca(II), due to the ionic charge and the strength of the corresponding bonds [61]. Therefore, any change in the $a/c$ ratio with ontogenetic development, as preliminarily observed in this study, has the potential to impact preferential uptake of cations during growth. This trend has implications for elemental concentrations obtained for palaeoseawater reconstruction and certainly warrants further investigation through elemental analysis and more unit cell parameter data.

The presence of organic matrix in crystal domains may cause misalignment of mosaic blocks, thus increasing mosaicity beyond what is observed in purely inorganic apatite minerals [64]. Moreover, biomineralization processes and the appositional growth of conodont bioapatite differ from crystal growth of natural apatite creating differences in crystalline structure. Two clear patterns of mosaicity are evident in the data presented herein (figure 5) identifying increasing mosaicity (disordering) of the single-crystal domains. Disorder increases both from tip to base in single specimens, and most importantly, disorder increases with ontogenetic development. The increased disorder with growth is possibly the result of the growth discontinuities between layer apposition and the increased potential to have mismatched crystal domains at these discontinuities. Defects in the alignment at a more juvenile stage could be propagated further with each new layer added, effectively increasing the degree of disorder with successive stages of development. Regardless of the cause, the potential significance of this disordering lies in the introduction of imperfections in either the lattices of individual nanocrystallites themselves or the ordering of nanocrystallites within each successive layer of crown tissue. Given that elemental uptake and trapping in biogenic apatite is the result of either adsorption or substitution, the degree of disordering has the potential to influence the degree to which a conodont may take up elements from seawater through either process. Therefore, the variation in disordering (mosaicity) between crown tissue types, as well as between element types ($S$, $M$, etc.), has the potential to influence the integrity of conodont bioapatite as a palaeoceanographic tracer. Care must be taken when comparing elemental concentrations of different tissue types [14], but based on the data presented here, it is likely that care must also be taken when comparing chemical concentrations of different apparatus elements within a single species.

# 5. Conclusion

Conodont bioapatite is an attractive target as a recorder of ancient seawater chemistry. At present, however, much remains unknown about the controls on elemental uptake, diagenesis and the ultimate integrity of conodont bioapatite as a seawater proxy. Whereas clear differences in the elemental concentrations of different tissue types have already been identified [14], the ultimate causes and consequences of these differences remain enigmatic. The systematic study of a single species presented here that used a novel approach to both X-ray beam size and sample selection demonstrated variations in crystallography and microstructure due to both ontogeny and element type within the conodont feeding apparatus. These results provide new support for the importance of functional adaptation in the microstructure of conodonts, as well as new insights into the possible mechanisms behind the variations in elemental concentration of different crown tissue types. Ontogeny and elemental position clearly impact crystallographic microstructure, and further systematic geochemical studies that directly consider phylogenetic relationships and functional morphology are required.

Data accessibility. Unit cell values and statistical analysis of the data can be found in the electronic supplementary material. Diffraction data are deposited in the Dryad Digital Repository: https://dx.doi.org/10.5061/dryad.2rbnzs7jn.
Authors' contributions. T.Z.F. and B.D.C. conceived of the study. N.E.B.M. processed the core and recovered the conodont samples used. M.S. generated all of the data presented here and drafted the initial version of the manuscript; all authors contributed to the writing and editing of the final version of the manuscript.

Competing interests. We declare we have no competing interests.

Funding. Diffraction studies by M.S. and T.Z.F. were supported by funding from the American Chemical Society-Petroleum Research Fund (ACS-PRF) grant no. ND-18485100. Conodont extraction and preparation was supported by United States National Science Foundation grant no. CAREER-1455030 to B.D.C.

Acknowledgements. We thank Dr Bruce Noll from Bruker AXS for his advice regarding D8 Quest instrument calibration and Debye ring analysis. We are also grateful to Prof. Alejandro Rodriguez-Navarro (Crystallography and Mineralogy, University of Granada, Spain) who provided access and advice for using the XRD2DScan software.

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
