## [Reviewer comments · Royal Society Open Science]

Review History

RSOS-200322.R0 (Original submission)

Review form: Reviewer 1

Is the manuscript scientifically sound in its present form?

No

Are the interpretations and conclusions justified by the results?

No

Is the language acceptable?

Yes

Do you have any ethical concerns with this paper?

No

Have you any concerns about statistical analyses in this paper?

Yes

Recommendation?

Major revision is needed (please make suggestions in comments)

Comments to the Author(s)

Comments on Shohel et al RSOS 2020

This is an interesting piece of work with original analyses and results that will be of interest to conodont specialists. However, the ms needs significant revision to better place it in context and take previous work into account. I also raise a number of issues that must be taken into account. The most significant are:

1. I do not think the material and what we know about the nature of *Dapsilodus* allows the issue of change through ontogeny to be addressed.
2. The 'zones' in elements cannot be compared in the way they are currently because the tips of elements of different sizes are not homologous.

Regarding XRD, I know a little about the basics of XRD but I am not able to comment on the technical aspect relating to XRD. My view is that such an evaluation is essential before a decision on publication is made.

Detailed comments are provided on the annotated pdf (Appendix A). These comments are also listed below by page and line.

I hope the authors are able to address my concerns and I look forward to seeing a revised version published in future.

Page: 4

Line 38

While this is true, it rather underplays some of what is already known. For example, Trotter and Eggins 2006 (already cited elsewhere in this ms) noted differences in susceptibility of different conodonts hard tissues to post mortem chemical alteration and demonstrated that white matter is more resistant. previous work should be properly acknowledged, and novelty of current work should not be overstated (there is novelty; putting it in context does not diminish this). Similarly previous, albeit preliminary work using EBSD to investigate conodont crystallinity is not mentioned in this ms (Perez-Huerta et al *Lethaia* 2012)

Line 43

standardizing terminology at this point in the ms would help the reader, esp if it aligns with general usage. The modern paleobiological/phylogenetic/histological conodont literature generally uses 'lamellar crown tissue' and 'white matter'. 'albid' and 'hyaline' are generally redundant and a little archaic. (I also note that in the glossary of the conodont treatise - admittedly a little old now - albid and hyaline refer to whole elements, not tissues. I suggest 'lamellar crown tissue' and 'white matter' throughout

Line 45

Inserted Text composed

Page: 5

Line 49

Donoghue and Purnell 1995 *Geology* worth citing here; Shirley et al essentially corroborated the conclusions of this work.

Line 50

This seems a rather harsh characterisation which implies previous studies were poorly designed. The reality in most cases is that they were designed to answer a question that is different to that upon which this ms focusses.

Page: 6

Line 76

I find this problematic. Because no articulated skeletons of *Dapsilodus* are known, there is no evidence that allows determination of what size elements occurred together in an apparatus. Furthermore, as noted below, we don't actually know how many of each element morphotype occurred in the apparatus, and whether duplicate elements of similar morphotype were of similar size. I do not think, given these uncertainties, that the authors can meaningfully talk about ontogenetic variation. They are talking about variation with element size and robustness. A reasonable case could be made that, for a particular element morphotype, this is likely to reflect ontogeny to some degree. However, different element types cannot be compared, and the Sb-c category could hide elements from multiple different locations. In the absence of fossils preserving an articulated apparatus, these uncertainties cannot be addressed, and I suggest the authors recast all aspects of this ms relating to 'ontogeny' in this light. I don't think elements of *dapsilodus* can meaningfully be classified as 'juvenile' and 'older'.
Wear is a red herring here, because from what we know of conodont development through life (noting that *dapsilodus* hasn't been studied, as far as I know). Multiple cycles of growth, use and wear, mean that wear can occur in elements of different sizes and ages (see Donoghue and Purnell 1999, *Geology*)

Line 79

This is a pretty poor quality figure. As a minimum, given that one of its primary purposes is to demonstrate size variation between classes, all elements should be shown at the same magnification. Aesthetic considerations are more subjective, but to my eye this is poorly put together and scruffy looking.

Page: 7

Line 87

Counts and ratios of different element morphotypes in collections of isolated elements are known to be a very unreliable guide to the numbers of elements in an articulated apparatus. As far as I know, in all cases where hypothetical ratios of isolated element types have been compared to the reality of actual articulated apparatuses, the hypothetical ratios have been shown to be incorrect. Although not explicitly addressing ratios in coniform conodonts, Purnell and Donoghue 2005 (*Special papers in Palaeontology*) discuss the biases.

Line 93

I find this a little confusing. Is a sample a spot on an element or an element? 'sample' and 'element' seem to be used almost, but not quite, interchangeably.
More importantly, these zones are later considered comparable (i.e. zones 1-3 in 'juvenile' elements are compared with zone 1-3 in 'older' elements). Leaving aside the question of whether these categories reflect ontogeny, these zones are not comparable in this way because of the way conodont elements add lamellae. They do not grow by basal addition, but by appositional growth, so the tip of a smaller/'juvenile' element is not homologous with the tip of a larger/'older' element, because the earlier growth stage is subsumed within the later one. The only comparable point - the only landmark that allows 'zones' in one element to be compared to those in another, is the tip of the basal cavity. Otherwise comparisons between elements of different sizes are meaningless. The least comparable parts of elements of different sizes are the tips.

Line 94

implies that not all analyses were conducted in the same way

Page: 8

Line 106

these seem to be critical to results and interpretation, but are not explained in a way that someone whose expertise is in conodonts would understand. Presumably, this is the target audience, so an explanation would be worthwhile.

Line 115

methods do not discuss statistics and statistical methods. e.g. normality of distributions, appropriateness of parametric tests, methods for regression fitting, approaches to significance testing (alpha values etc).

Line 119

I found it difficult to see the patterns that the authors describe. Non-expert readers not used to interpreting this type of data will need a more detailed description if they are to follow how the patterns in the data support the interpretations presented.

I struggled to see the distinct differences highlighted by the authors below; without this it's hard to judge the degree to which the data support the interpretations, and this should not be a matter of trust.

Presumably, as whole elements were analysed, the X-ray beam passes through surface layers of lamellar crown tissue before it reaches the white matter. If so, do the patterns from white matter zones reflect a signal that mixes white matter and lamellar crown?

Page 11

Fig 2 caption:

I have no experience of interpreting plots like this, but the images themselves seem to lack contrast, which makes it hard to pick out patterns clearly.

Could contrast be enhanced, or is absolute intensity part of the signal?

Page 12

Line 137

This seems to be methods

Page: 14

Figure 3 caption

the combining of data in these plots and the statistical tests reported as if they represent categories that are directly comparable is wrong, for the reasons outlined above. 'Older' elements of different morphotypes (M, Sa, and Sb-c) cannot meaningfully be combined. Same applies to 'juvenile'.

Also, the plots show what I assume are regressions. these are problematic. Firstly, there is no indication of how well they represent the data. more importantly, the text suggests that there is no reason to choose a or c as dependent or independent variable, and both are equally prone to measurement error. in which case simple least squares linear regression is not appropriate. reduced major axis or orthogonal regression should be used.

no P value reported

no P value reported

Page: 15

Line 161-164

this mode of reporting looks odd; is it intended to imply an alpha value of 0.05 as a threshold for significance?

Line 165

no P values

Page: 16

Fig 4 caption

as noted above, zones are not comparable in this way; tips are not homologous in 'older' and 'juvenile' elements.

Regarding b, I'm not sure that the diagram is an adequate representation of increasing mosaicity as it implies disruption rather than variation in crystallite orientation

Line 192

an expanded version of this section, earlier in the ms, would help the reader to understand the results

Page: 18

Line 230

more detailed subsequent work perhaps also worth citing - e.g. Martinez-Perez et al Palaeontology 2014.

Page: 20

Line 264

somewhere, whether in the introduction or here, the authors need to acknowledge that the variation in $\delta^{18}\text{O}$ within conodonts and between elements in the same taxon, has been investigated directly. Wheeley et al 2012 (J Geol Soc London) found minimal difference between white matter and lamellar crown tissue. It thus appears that the conclusion drawn here, that crystallographic differences are likely to have an impact of geochemical signals in conodont elements, is unsupported by available data.

Line 276

this is not how conodont elements grow. they grow through distinct phases, with prolonged period of function between. defects are not propagated between phases. See Donoghue and Purnell 1999 Geology, and Shirley et al 2018 (already cited)

Page 21

Line 287

Addressed by Wheeley et al 2012 J Geol Soc London

Review form: Reviewer 2

Is the manuscript scientifically sound in its present form?

No

Are the interpretations and conclusions justified by the results?

No

Is the language acceptable?

Yes

Do you have any ethical concerns with this paper?

No

Have you any concerns about statistical analyses in this paper?

Yes

Recommendation?

Reject

Comments to the Author(s)

It is a very interesting study and I would absolutely encourage publication of the data. The manuscript, in its current form, has several flaws, which - taken together - mean that a complete rewriting is needed to address the limitations of the analytical method and the material (taxon choice and histology), as well as correcting statistical analyses. There is only a handful of studies on this topic in total and still a few of them are not cited or mis-cited. A better integration and discussion with previous studies is needed and it will likely change the interpretation and conclusions substantially.

METHODS

The method employed in the study allows in situ X-ray diffraction and deriving unit cell parameters and subsequent crystallographic parameters such as mosaicity from the diffraction of an X-ray beam on the surface of the specimen. Diffraction and effective analysed area is affected by sample topography and quantitative unit cell analyses should be performed on flat, polished surfaces to minimise diffraction artifacts. Conodont element surface has curvature that is substantial compared to the spot size used (120 μm). Moreover, the study makes a point about crystallographic parameters changing along the element with lower mosaicity towards the cusp. This ignores the effect of histology: the measurement effectively compares two types of tissues which are known already to consist of crystals with largely differing sizes and ordering. The cusp consists of the albid tissue which, going down the element, is increasingly embedded in hyaline tissue. The difference in mosaicity presented in the study is not meaningful if these systematic differences of tissues are not accounted for. This is virtually impossible when measuring element surface, which will provide a mixture of values for both tissues, and would require a sectioned specimen. This is the most likely explanation for lower polycrystallinity of M elements compared to others: this might be simply due to a higher proportion of albid tissue. The same limitation applies to the proposed observation of increasing a-axis/c-axis ratio during growth: the authors attribute this to a change in chemical composition, but do not compare it with actual data, which is available (e.g. Trotter et al. 2007), and do not consider differences between tissues, which are the most likely explanation.

The authors consistently refer to areas with low mosaicity as “single crystal” or “single crystalline”, which is very confusing, since they themselves document that they consist of multiple nanocrystals. A more appropriate term would be needed.

Preferred crystallographic orientations are not meaningful if they are not provided with respect to sample orientation (lines 192-206).

STATE OF KNOWLEDGE AND DISCUSSION

There is quite some misattribution of data in the citations, which is partially due to a conceptual mixup of nanocrystals and crystallites. For example, references 20 and 21 are cited as the source of information about the size of nanocrystals in conodont tissue, whereas they do not describe or even mention nanocrystals at all. The methods used in these studies are not capable of discerning nanocrystals. The studies which employed appropriate methods and yielded this type of data are not mentioned at all (Pietzner et al. 1968; Nemliher & Kallaste 2012; or in articles cited elsewhere in the manuscript but not in this context: Ferretti et al 2017). Taking these studies into account would substantially change the interpretations and the formulation of the study question.

Lines 209-210: “albid crown tissue is made of large compact crystals arranged along the c-axes parallel or sub-parallel to the long axis of the element” - the long axis of a coniform element is not the same as the long axis of e.g. pectiniform element, therefore this citation cannot be automatically transposed onto current study; considering the geometry of the elements, it would be c-axes perpendicular to the long axis of the studied taxon.

MATERIAL

The study attempts to link measured parameters with conodont functional morphology, but the chosen conodont is nearly the worst possible taxon for the task, as it has no apparatus reconstruction in the sense of positional homology and element types as referred to in the

manuscript (Sa, Sb-c, M) do not express homology and therefore cannot be linked to functional models in better understood conodonts even by analogy. The only functional model available for coniform conodonts, Murdock et al. 2013 10.1098/rspb.2013.1524, is not mentioned, although it would provide criteria for the interpretation. The mechanics of coniform conodonts, and of *Dapsilodus* of all of them, are the least understood of all euconodonts. Given that the method is not destructive, it is confusing that the authors did not address a taxon with an apparatus reconstruction and an occlusal model, or at least one that wouldn't cause the issues with tissue differentiation, e.g. a younger taxon with reduced proportions of albid matter.

The criteria for identifying ontogenetic stage wouldn't be reproducible for other researchers: "ontogenetic development was determined by appearance (size, robustness, apparent wear, transparency)": size is very hard to assess for the reader as the elements in Fig 1 are not scaled to the same magnification; apparent wear is known not to be reliable given that conodonts periodically grew new tissue throughout their life; transparency is mostly due to peculiarities of preservation and is known to vary from sample to sample; robustness is rather vague altogether). Some quantitative measure of ontogenetic stage (size, if nothing better is available) should be introduced.

Decision letter (RSOS-200322.R0)

12-May-2020

Dear Mr Shohel,

The editors assigned to your paper ("Ontogenetic variability in crystallography and mosaicism of conodont apatite: Implications for microstructure, paleothermometry and geochemistry") have now received comments from reviewers. We would like you to revise your paper in accordance with the referee and Associate Editor suggestions which can be found below (not including confidential reports to the Editor). Please note this decision does not guarantee eventual acceptance.

Please submit a copy of your revised paper before 04-Jun-2020. Please note that the revision deadline will expire at 00.00am on this date. If we do not hear from you within this time then it will be assumed that the paper has been withdrawn. In exceptional circumstances, extensions may be possible if agreed with the Editorial Office in advance. We do not allow multiple rounds of revision so we urge you to make every effort to fully address all of the comments at this stage. If deemed necessary by the Editors, your manuscript will be sent back to one or more of the original reviewers for assessment. If the original reviewers are not available, we may invite new reviewers.

- Data accessibility

If you wish to submit your supporting data or code to Dryad (<http://datadryad.org/>), or modify your current submission to dryad, please use the following link:
<http://datadryad.org/submit?journalID=RSOS&manu=RSOS-200322>

- Competing interests

- Authors' contributions

- Acknowledgements

- Funding statement

on behalf of Professor Rachel Wood (Associate Editor)
openscience@royalsociety.org

Associate Editor's comments (Professor Rachel Wood):

Associate Editor: 1

Comments to the Author:

Both reviewers comment that they found this is a very interesting study. Both raise, however, major and diverse issues that must be addressed before publication can be considered.

Comments to Author:

Reviewers' Comments to Author:

Reviewer: 1

Comments to the Author(s)

Comments on Shohel et al RSOS 2020

This is an interesting piece of work with original analyses and results that will be of interest to conodont specialists. However, the ms needs significant revision to better place it in context and take previous work into account. I also raise a number of issues that must be taken into account. The most significant are:

1. I do not think the material and what we know about the nature of *Dapsilodus* allows the issue of change through ontogeny to be addressed.
2. The 'zones' in elements cannot be compared in the way they are currently because the tips of elements of different sizes are not homologous.

Regarding XRD, I know a little about the basics of XRD but I am not able to comment on the technical aspect relating to XRD. My view is that such an evaluation is essential before a decision on publication is made.

Detailed comments are provided on the annotated pdf. These comments are also listed below by page and line.

I hope the authors are able to address my concerns and I look forward to seeing a revised version published in future.

Page: 4

Line 38

While this is true, it rather underplays some of what is already known. For example, Trotter and Eggins 2006 (already cited elsewhere in this ms) noted differences in susceptibility of different conodonts hard tissues to post mortem chemical alteration and demonstrated that white matter is more resistant. previous work should be properly acknowledged, and novelty of current work should not be overstated (there is novelty; putting it in context does not diminish this). Similarly

previous, albeit preliminary work using EBSD to investigate conodont crystallinity is not mentioned in this ms (Perez-Huerta et al Lethaia 2012)

Line 43

standardizing terminology at this point in the ms would help the reader, esp if it aligns with general usage. The modern paleobiological/phylogenetic/histological conodont literature generally uses 'lamellar crown tissue' and 'white matter'. 'albid' and 'hyaline' are generally redundant and a little archaic. (I also note that in the glossary of the conodont treatise - admittedly a little old now - albid and hyaline refer to whole elements, not tissues. I suggest 'lamellar crown tissue' and 'white matter' throughout

Line 45

Inserted Text composed

Page: 5

Line 49

Donoghue and Purnell 1995 Geology worth citing here; Shirley et al essentially corroborated the conclusions of this work.

Line 50

This seems a rather harsh characterisation which implies previous studies were poorly designed. The reality in most cases is that they were designed to answer a question that is different to that upon which this ms focusses.

Page: 6

Line 76

I find this problematic. Because no articulated skeletons of *Dapsilodus* are known, there is no evidence that allows determination of what size elements occurred together in an apparatus. Furthermore, as noted below, we don't actually know how many of each element morphotype occurred in the apparatus, and whether duplicate elements of similar morphotype were of similar size. I do not think, given these uncertainties, that the authors can meaningfully talk about ontogenetic variation. They are talking about variation with element size and robustness. A reasonable case could be made that, for a particular element morphotype, this is likely to reflect ontogeny to some degree. However, different element types cannot be compared, and the Sb-c category could hide elements from multiple different locations. In the absence of fossils preserving an articulated apparatus, these uncertainties cannot be addressed, and I suggest the authors recast all aspects of this ms relating to 'ontogeny' in this light. I don't think elements of *dapsilodus* can meaningfully be classified as 'juvenile' and 'older'.

Wear is a red herring here, because from what we know of conodont development through life (noting that *dapsilodus* hasn't been studied, as far as I know). Multiple cycles of growth, use and wear, mean that wear can occur in elements of different sizes and ages (see Donoghue and Purnell 1999, Geology)

Line 79

This is a pretty poor quality figure. As a minimum, given that one of its primary purposes is to demonstrate size variation between classes, all elements should be shown at the same magnification. Aesthetic consideration are more subjective, but to my eye this is poorly put together and scruffy looking.

Page: 7

Line 87

Counts and ratios of different element morphotypes in collections of isolated elements are known to be a very unreliable guide to the numbers of elements in an articulated apparatus. As far as I know, in all cases where hypothetical ratios of isolated element types have been compared to the

reality of actual articulated apparatuses, the hypothetical ratios have been shown to be incorrect. Although not explicitly addressing ratios in coniform conodonts, Purnell and Donoghue 2005 (Special papers in Palaeontology) discuss the biases.

Line 93

I find this a little confusing. is a sample a spot on an element or an element? 'sample' and 'element' seem to be used almost, but not quite, interchangeably.

More importantly, these zones are later considered comparable (i.e. zones 1-3 in 'juvenile' elements are compared with zone 1-3 in 'older' elements. Leaving aside the question of whether these categories reflect ontogeny, these zones are not comparable in this way because of the way conodont elements add lamellae. They do not grow by basal addition, but by appositional growth, so the tip of a smaller/'juvenile' element is not homologous with the tip of a larger/'older' element, because the earlier growth stage is subsumed within the later one. The only comparable point - the only landmark that allows 'zones' in one element to be compared to those in another, is the tip of the basal cavity. otherwise comparisons between elements of different sizes are meaningless. The least comparable parts of elements of different sizes are the tips.

Line 94

implies that not all analysis were conducted in the same way

Page: 8

Line 106

these seem to be critical to results and interpretation, but are not explained in a way that someone whose expertise is in conodonts would understand. Presumably, this is the target audience, so an explanation would be worthwhile.

Line 115

methods do not discuss statistics and statistical methods. e.g. normality of distributions, appropriateness of parametric tests, methods for regression fitting, approaches to significance testing (alpha values etc).

Line 119

I found it difficult to see the patterns that the authors describe. Non-expert readers not used to interpreting this type of data will need a more detailed description if they are to follow how the patterns in the data support the interpretations presented.

I struggled to see the distinct differences highlighted by the authors below; without this its hard to judge the degree to which the data support the interpretations, and this should not be a matter of trust.

Presumably, as whole elements were analysed, the X-ray beam passes through surface layers of lamellar crown tissue before it reaches the white matter. If so, do the patterns from white matter zones reflect a signal that mixes white matter and lamellar crown?

Page 11

Fig 2 caption:

I have no experience of interpreting plots like this, but the images themselves seem to lack contrast, which makes it hard to pick out patterns clearly.

Could contrast be enhanced, or is absolute intensity part of the signal?

Page 12

Line 137

This seems to be methods

Page: 14

Figure 3 caption

the combining of data in these plots and the statistical tests reported as if they represent categories that are directly comparable is wrong, for the reasons outlined above. 'Older' elements of different morphotypes (M, Sa, and Sb-c) cannot meaningfully be combined. Same applies to 'juvenile'.

Also, the plots show what I assume are regressions. these are problematic. Firstly, there is no indication of how well they represent the data. more importantly, the text suggests that there is no reason to chose a or c as dependent or independent variable, and both are equally prone to measurement error. in which case simple least squares linear regression is not appropriate. reduced major axis or orthogonal regression should be used.

no P value reported

no P value reported

Page: 15

Line 161-164

this mode of reporting looks odd; is it intended to imply an alpha value of 0.05 as a threshold for significance?

Line 165

no P values

Page: 16

Fig 4 caption

as noted above, zones are not comparable in this way; tips are not homologous in 'older' and 'juvenile' elements.

Regarding b, I'm not sure that the diagram is an adequate representation of increasing mosaicity as it implies disruption rather than variation in crystallite orientation

Line 192

an expanded version of this section, earlier in the ms, would help the reader to understand the results

Page: 18

Line 230

more detailed subsequent work perhaps also worth citing - e.g. Martinez-Perez et al Palaeontology 2014.

Page: 20

Line 264

somewhere, whether in the introduction or here, the authors need to acknowledge that the variation in delta 18 O within conodonts and between elements in the same taxon, has been investigated directly. Wheeley et al 2012 (J Geol Soc London) found minimal difference between white matter and lamellar crown tissue. It thus appears that the conclusion drawn here, that crystallographic differences are likely to have an impact of geochemical signals in conodont elements, is unsupported by available data.

Line 276

this is not how conodont elements grow. they grow through distinct phases, with prolonged period of function between. defects are not propagated between phases. See Donoghue and Purnell 1999 Geology, and Shirley et al 2018 (already cited)

Page 21

Line 287

Addressed by Wheeley et al 2012 J Geol Soc London

Reviewer: 2

Comments to the Author(s)

It is a very interesting study and I would absolutely encourage publication of the data. The manuscript, in its current form, has several flaws, which - taken together - mean that a complete rewriting is needed to address the limitations of the analytical method and the material (taxon choice and histology), as well as correcting statistical analyses. There is only a handful of studies on this topic in total and still a few of them are not cited or mis-cited. A better integration and discussion with previous studies is needed and it will likely change the interpretation and conclusions substantially.

METHODS

The method employed in the study allows in situ X-ray diffraction and deriving unit cell parameters and subsequent crystallographic parameters such as mosaicity from the diffraction of an X-ray beam on the surface of the specimen. Diffraction and effective analysed area is affected by sample topography and quantitative unit cell analyses should be performed on flat, polished surfaces to minimise diffraction artifacts. Conodont element surface has curvature that is substantial compared to the spot size used (120 μm). Moreover, the study makes a point about crystallographic parameters changing along the element with lower mosaicity towards the cusp. This ignores the effect of histology: the measurement effectively compares two types of tissues which are known already to consist of crystals with largely differing sizes and ordering. The cusp consists of the albid tissue which, going down the element, is increasingly embedded in hyaline tissue. The difference in mosaicity presented in the study is not meaningful if these systematic differences of tissues are not accounted for. This is virtually impossible when measuring element surface, which will provide a mixture of values for both tissues, and would require a sectioned specimen. This is the most likely explanation for lower polycrystallinity of M elements compared to others: this might be simply due to a higher proportion of albid tissue. The same limitation applies to the proposed observation of increasing a-axis/c-axis ratio during growth: the authors attribute this to a change in chemical composition, but do not compare it with actual data, which is available (e.g. Trotter et al. 2007), and do not consider differences between tissues, which are the most likely explanation.

The authors consistently refer to areas with low mosaicity as “single crystal” or “single crystalline”, which is very confusing, since they themselves document that they consist of multiple nanocrystals. A more appropriate term would be needed.

Preferred crystallographic orientations are not meaningful if they are not provided with respect to sample orientation (lines 192-206).

STATE OF KNOWLEDGE AND DISCUSSION

There is quite some misattribution of data in the citations, which is partially due to a conceptual mixup of nanocrystals and crystallites. For example, references 20 and 21 are cited as the source of information about the size of nanocrystals in conodont tissue, whereas they do not describe or even mention nanocrystals at all. The methods used in these studies are not capable of discerning nanocrystals. The studies which employed appropriate methods and yielded this type of data are not mentioned at all (Pietzner et al. 1968; Nemliher & Kallaste 2012; or in articles cited elsewhere in the manuscript but not in this context: Ferretti et al 2017). Taking these studies into account would substantially change the interpretations and the formulation of the study question.

Lines 209-210: “albid crown tissue is made of large compact crystals arranged along the c-axes parallel or sub-parallel to the long axis of the element” - the long axis of a coniform element is not the same as the long axis of e.g. pectiniform element, therefore this citation cannot be automatically transposed onto current study; considering the geometry of the elements, it would be c-axes perpendicular to the long axis of the studied taxon.

MATERIAL

The study attempts to link measured parameters with conodont functional morphology, but the chosen conodont is nearly the worst possible taxon for the task, as it has no apparatus reconstruction in the sense of positional homology and element types as referred to in the manuscript (Sa, Sb-c, M) do not express homology and therefore cannot be linked to functional models in better understood conodonts even by analogy. The only functional model available for coniform conodonts, Murdock et al. 2013 10.1098/rspb.2013.1524, is not mentioned, although it would provide criteria for the interpretation. The mechanics of coniform conodonts, and of *Dapsilodus* of all of them, are the least understood of all euconodonts. Given that the method is not destructive, it is confusing that the authors did not address a taxon with an apparatus reconstruction and an occlusal model, or at least one that wouldn't cause the issues with tissue differentiation, e.g. a younger taxon with reduced proportions of albid matter.

The criteria for identifying ontogenetic stage wouldn't be reproducible for other researchers: "ontogenetic development was determined by appearance (size, robustness, apparent wear, transparency)": size is very hard to assess for the reader as the elements in Fig 1 are not scaled to the same magnification; apparent wear is known not to be reliable given that conodonts periodically grew new tissue throughout their life; transparency is mostly due to peculiarities of preservation and is known to vary from sample to sample; robustness is rather vague altogether). Some quantitative measure of ontogenetic stage (size, if nothing better is available) should be introduced.

Author's Response to Decision Letter for (RSOS-200322.R0)

See Appendix B.

Decision letter (RSOS-200322.R1)

13-Jun-2020

Dear Mr Shohel,

It is a pleasure to accept your manuscript entitled "Ontogenetic variability in crystallography and mosaicism of conodont apatite: Implications for microstructure, paleothermometry and geochemistry" in its current form for publication in Royal Society Open Science.

on behalf of Professor Rachel Wood (Associate Editor)
openscience@royalsociety.org

Appendix A**ROYAL SOCIETY
OPEN SCIENCE****Ontogenetic variability in crystallography and mosaicity of
conodont apatite: Implications for microstructure,
paleothermometry and geochemistry**

Journal:	Royal Society Open Science
Manuscript ID	RSOS-200322
Article Type:	Research
Date Submitted by the Author:	28-Feb-2020
Complete List of Authors:	Shohel, Mohammad; University of Iowa, Department of Chemistry McAdams, Neo; Texas Tech University, Department of Geosciences Cramer, Bradley; University of Iowa, Department of Earth and Environmental Sciences Forbes, Tori; University of Iowa, Department of Chemistry
Subject:	Palaeontology < EARTH SCIENCES
Keywords:	Conodont, Ontogeny, Microstructure, Diffraction, Mosaicity, Texturing
Subject Category:	Earth and Environmental Science

Author-supplied statements

Relevant information will appear here if provided.

Ethics

Does your article include research that required ethical approval or permits?:

This article does not present research with ethical considerations

Statement (if applicable):

CUST_IF_YES_ETHICS :No data available.

Data

It is a condition of publication that data, code and materials supporting your paper are made publicly available. Does your paper present new data?:

Yes

Statement (if applicable):

The diffraction frames are deposited in Dryad. It can be accessible through following secured link:

<https://datadryad.org/stash/share/MM3fY7Uy-iGwHCGPSHVqsU7qOxGkjTYbWhyIIXTYL5k>

Conflict of interest

I/We declare we have no competing interests

Statement (if applicable):

CUST_STATE_CONFLICT :No data available.

Authors' contributions

This paper has multiple authors and our individual contributions were as below

Statement (if applicable):

T.Z.F. and B.D.C. conceived of the study. N.E.B.M. processed the core and recovered the conodont samples utilized. M.S. generated all of the data presented here and drafted the initial version of the manuscript; all authors contributed to the writing and editing of the final version of the manuscript.

**Ontogenetic variability in crystallography and mosaicity of conodont apatite: Implications**
**for microstructure, paleothermometry and geochemistry**

Mohammad Shohel¹, Neo E.B. McAdams², Bradley D. Cramer^{*3}, Tori Z. Forbes^{*1}

¹Department of Chemistry, University of Iowa, Iowa City, IA 52242, USA

²Department of Geosciences, Texas Tech University, Lubbock, TX 79409, USA

³Department of Earth and Environmental Sciences, University of Iowa, Iowa City, IA 52242,
USA

*Corresponding authors: tori-forbes@uiowa.edu; bradley-cramer@uiowa.edu

**Abstract:**

X-ray diffraction data from Silurian conodonts belonging to various developmental stages of the
species *Dapsilodus obliquicostatus* demonstrate changes in crystallography and degree of
nanocrystallite ordering (mosaicity) in both hyaline and albid crown tissue. The exclusive use of a
single species in this study, combined with systematic testing of each element type at multiple
locations, provided insight into microstructural and crystallographic differentiation between
element position (S_a , S_{b-c} , M) as well as between juveniles and adults. A relative increase in the
unit cell dimensions of the a -axis/ c -axis ratio of nanocrystallites during growth was apparent in
areas demonstrating single-crystal behavior but no such relationship was seen in dominantly
polycrystalline areas. Systematic variations in mosaicity were identified, with mosaicity (as a
proxy for disorder) increasing during growth, as well as along elements from tip to base. These
results provide potential insight into the integrity of conodont apatite as a recorder of

paleoseawater chemistry, as well as demonstrate the need to consider the influence of ontogeny
and element position on the use of conodonts in paleothermometry and geochemical investigations.

**Keywords**

Conodont, Ontogeny, Microstructure, Diffraction, Mosaicity, Texturing

**1. Introduction**

Bioapatites from conodont microfossils are routinely utilized to develop basin-scale thermal
histories [1], for paleothermometry of the ocean-atmosphere system [2-5], and are increasingly
utilized as a proxy for ancient ocean chemistry [6-14]. They are particularly useful given their
biostratigraphic significance and nearly ubiquitous presence in marine strata, and numerous studies
have illustrated that conodont apatite is more resistant to diagenetic alteration than is brachiopod
calcite [4,5,15]. Even seemingly pristine brachiopods screened using established trace element
discrimination methods can be geochemically altered [4,15,16], which has promoted the use of
conodont bioapatite for reconstruction of paleoseawater chemistry. However, much remains
unknown about conodont bioapatite formation processes, potential vital effects on elemental and
isotopic compositions, the role of microstructure on chemical retention, and ontogenetic variations
in crystal chemistry.

Early studies identified two distinct types of crown tissue in conodonts and a less
commonly preserved distinct tissue type of the basal body [17,18]. The translucent hyaline, and
typically opaque, white-matter albid comprise the crown tissue of euconodonts [19-21]. All tissue
types are comprised of nanocrystals of apatite ranging from 10s to ~100 nm wide by 0.1 to 10 μm
long that are embedded in an organic matrix [20-22]. The ratio of nanocrystal to organic matrix

varies between the three tissue types with albid crown tissue typically being the most crystal rich
(or behaving as a single crystal) and containing the least amount of organic matrix. Whereas the
significance of phylogeny and ontogeny in microstructure has been well documented [21,53],
nearly all modern studies of conodont crystallography have chosen to combine crystallographic
data from different species, different element types, different intervals of Earth history, and
different stages of ontogenetic development [39,47,52]. The net result has often been aggregated
data that show no significant trends and a conclusion that conodont bioapatite crystallography is
essentially identical across the clade. Here, we focus on a single species of coniform-bearing
conodont, *Dapsilodus obliquicostatus* (Branson and Mehl, 1933) [23], by utilizing two-
dimensional micro X-ray diffraction techniques that allowed us to examine discrete portions of
each conodont element. We examined 14 specimens and analyzed 63 total spots through micro X-
ray diffraction from multiple positions along each specimen. By systematically evaluating the
differences between element type (S_a , S_{b-c} , M), as well as the differences between growth stages
of each element type, we can address the role of ontogeny in both crystallography as well as the
structural ordering (mosaicity) of nanocrystallites within conodont bioapatite.

**2. Materials and Methods**

Conodont samples utilized in this study came from the Schlamer #1 drillcore, Alexander County,
southwestern Illinois, USA [24]. All specimens analyzed were recovered from the St. Clair
Formation and are well preserved and thermally unaltered with a conodont colour alteration index
(CAI) of 1 indicating a burial temperature no higher than 80°C [1]. The low CAI of these
specimens demonstrates a very low likelihood of pyrolysis of the original organic matter in the
bioapatite due to metamorphism. The St. Clair Formation records the Wenlock Epoch of the

Silurian Period and spans an interval from approximately 433 to 426.75 Ma [24,25]. Carbonate
and carbonaceous shale samples from the core ranged in size from 7.5 cm to 15cm in stratigraphic
thickness and were digested in the University of Iowa Micropaleontology Laboratory using the
standard double-buffered formic acid technique [26]. Insoluble residues were further processed by
heavy liquid separation utilizing lithium metatungstate (LMT) at a density of 2.83-2.84 kg/L, and
the remaining heavy fraction was picked under binocular microscope for conodonts. Specimens of
*D. obliquicostatus* were selected to represent a range of ontogenetic development from juvenile to
gerontic (figure 1), and ontogenetic development was determined by appearance (size, robustness,
apparent wear, transparency).

**Figure 1.** Conodont elements of *Dapsilodus obliquicostatus* used in the present study. Each
scale bar corresponds to 100 μm .

The apparatus of *D. obliquicostatus* [27] includes an acodontiform M element, a
distacodontiform S_a element modified with lateral costae that occasionally extend behind the

posterior keel, and distacodontiform S_b and S_c elements that exhibit slight to strong twisting along
the cusp [28]. The degree of twisting is the basis for differentiation between S_b and S_c elements,
but given the lack of a precise definition combined with the change exhibited over time, most
authors tabulate a combined S_{b-c} element type [28] and we have followed this procedure herein.
The entire apparatus has not been found *in situ* but approximations of elements in the apparatus
based upon ratio counts indicates an apparatus with a ratio of $1S_a:10S_{b-c}:5M$ [24,27-29]. None of
the specimens included in this study included a preserved basal body.

X-ray diffraction experiments were carried out in a Bruker D8 Quest single crystal X-ray
diffractometer equipped with a CMOS area detector that allows diffraction pattern analysis of both
single-crystalline and polycrystalline material. Irradiation of samples occurred through a
microfocus Mo $K\alpha$ ($\lambda = 0.7107 \text{ \AA}$) X-ray source with a beam diameter of $120 \mu\text{m}$, which is suitable
to analyze multiple positions along each specimen. Depending on the size of the sample, three to
six zones were analyzed along the length of each element. In a typical experiment, conodont
elements were coated with mineral oil, then attached on top of MiTeGen Dual-Thickness
MicroLoop in a vertical position. The sample was placed on the three-circle goniometer and the
position of the X-ray beam was chosen using the microscope camera.

X-ray diffraction data associated with each position were collected in reflection mode by
varying the Φ angle while keeping the other angles fixed ($2\theta=0^\circ$, $\omega=0^\circ$ and $\chi=54.74^\circ$). An eight- to
eleven-second exposure time was needed for analyzing albid and hyaline material and a 20-30
second exposure time was needed for the basal cavity. All experiments were carried out at 100°K
to reduce thermal motion of atoms. For single crystalline regions, unit cell parameters were
determined identifying the well-resolved reflections within the 2D frames, followed by indexing
and refinement of the a and c parameters using Bruker APEX3 software. Mosaicity values

represent the full-width half-max of the reflections and are calculated automatically by the APEX3
software during the unit cell refinement process. For polycrystalline zones, the Debye rings were
integrated using APEX3 software to create the resulting powder X-ray diffraction (PXRD) pattern.
Unit cell parameters were obtained by indexing the PXRD pattern by the least square cell
parameter program [30]. XRD peaks that were well separated and did not overlap with other peaks
were selected for indexing. Alignment of the diffractometer was calibrated using the standard
YLID crystal provided by Bruker AXS. Detector distance, swing angle, and beam center were
calibrated by comparing the 2θ angles and detector position of a polycrystalline biomimetic
fluorapatite standard produced in the University of Iowa Department of Chemistry [31,32].
Preferred crystallographic orientation was analyzed by Psi scan (also known as Beta angle) using
XRD2DScan software [33].
31 117 **3. Results**

32 33 118 **(a) 2D X-ray diffraction and unit cell parameters**

The 2D diffraction patterns indicate variability of crystalline structure along the length of
*Dapsilodus obliquicostatus* elements (figure 2). For juvenile S_a and S_{b-c} elements, we observed
single and separated reflections in the diffraction pattern from both albid to hyaline, indicating an
ordered single crystalline apatite structure in both tissue types. In contrast, older S_a and S_{b-c}
elements displayed elongation and texturing of the individual reflections within both albid and
hyaline areas, indicating some level of disorder in coherent domains of diffraction. All M elements
possess single and separated reflections within both albid and hyaline regions despite their

(a)

(b)

Figure 2. Representative 2D X-ray diffraction patterns showing variation of crystalline structure at different zones of juvenile and older elements along the length (a) S_a (b) S_{b-c} (c) M. The basal cavity was polycrystalline as seen from Debye rings in the diffraction pattern of the last zone. Other zones in albid crown material were single crystalline with separate spots in the diffraction pattern. For older S_a and S_{b-c} elements, elongation of the reflections were observed in single crystalline zones.

ontogeny, indicating a highly-ordered crystalline structure and a clear differentiation from S
elements. The basal cavity of all conodont elements in our study are polycrystalline as evident by
the presence of Debye rings in 2D diffraction patterns, however, we left the basal cavity intact for
all samples (i.e. there was the open space of the cavity itself). Interference resulting from X-rays
passing through the outer hyaline portion of the basal cavity, into the open space of the cavity, and
into the opposite outer hyaline portion of the basal cavity may pose a question about the Debye
ring formation. Therefore, we ran an additional experiment on a subsample of bioapatite taken
from a basal cavity and also observed Debye ring formation, proving inherent polycrystallinity in
that zone.

We determined unit cell parameters using single crystalline or polycrystalline protocols
separately. For single crystalline regions, reflections were harvested to determine the reciprocal
lattice. Using Fourier transform, the reciprocal lattice was converted into direct lattice and then
unit cell parameters were determined. For polycrystalline regions, Debye rings in the diffraction
pattern were integrated, after which the d-spacing for each diffraction peak was calculated and
used to determine unit cell parameters. For some regions, unit cell parameters and mosaicity
(indicated with NR in Table S1, S2 and S3 of SI) could not be calculated due to (i) overlap of the
Debye ring and single crystal reflections, (ii) significant texturing of the Debye rings, which left
determination of unit cell parameters by either of our protocols unsuitable, or (iii) insufficient
number of diffraction spots.

Calculated unit cell parameters for single crystalline and polycrystalline regions of
conodont elements are shown in figure 3. The average unit cell parameter a for polycrystalline
zones was within error of single crystalline zones with values 9.385 ± 0.017 and 9.371 ± 0.018 Å,
respectively. The values were lower than the a value of Durango apatite and volcanic apatite

minerals analyzed in our study and other literature values (Table S4 in SI). A similar relationship
was observed for c with averages of 6.870 ± 0.021 for polycrystalline zones and 6.892 ± 0.015 Å
for single crystalline regions. No significant deviation or systematic variation in the value of each
axis individually was observed for samples from different ontogenetic stages or type of element.

Figure 3. Binary plots showing unit cell parameters of conodont elements refined from (a) single crystalline zones and (b) polycrystalline zones. Measured unit cell parameter of some apatite minerals are also in (a) and (b) for comparison. A positive Pearson's correlation coefficient was found in between a and c of single crystalline zones with $r= 0.5853$ and $r= 0.7201$ for juvenile and older elements, respectively. In contrast, no statistically significant relationship was found for polycrystalline zones with $r= -0.5689$ and $r= 0.3334$ for juvenile and older elements, respectively.

A positive Pearson's correlation coefficient with $r= 0.6466$ ($P<0.0001$, 95% confidence interval)
was found between a and c for all single crystalline zones (figure S1). However, rather stronger
correlation was observed for older elements with $r= 0.7201$ ($P<0.0001$, 95% confidence interval)
than the juveniles with $r= 0.5853$ ($P=0.0019$, 95% confidence interval) (figure 3). In contrast, no
statistical significance was found between a and c for all polycrystalline zone with $r= -0.5689$ and
$r= 0.3334$ for juvenile and older elements, respectively (figure 3b and figure S1). There is a clear
change in the a -axis/ c -axis ratio with respect to ontogeny in single crystalline zones (figure 3a)
due to a relative increase in the a -axis length compared to the c -axis in older specimens of all
element types.

**(b) Mosaicity and texturing**

Mosaicity describes the level of crystalline ordering observed within the material and can be used
as a metric describing that crystallinity. Perfect single crystals have no defects within the lattice
and unit cells can be described as an infinite three-dimensional tiling. However, most crystals are
not perfect and contain defects that then cause subtle misalignment of the stacked unit cells
(domain of coherent X-ray scattering). Therefore the mosaicity of the material describes the degree
of crystalline imperfections and allows us to describe the conodont sample as composed of mosaic
blocks with different levels of alignment [34]. The value of mosaicity we report is dependent on
the spread of reflections observed within the X-ray diffraction and described by evaluating the
full-width half-max of the reflection when evaluating the intensity versus rotation angle plot
[35,36].

The average mosaicity of conodont single crystalline zones was $1.26 \pm 0.07^\circ$, which was
higher than natural apatite minerals ($0.955 \pm 0.005^\circ$), suggesting crystalline imperfections due to

disorder (figure 4a and figure 4b). Moreover, the mosaicity of juvenile elements ($1.21 \pm 0.07^\circ$) was
 lower than

Figure 4. (a) Mosaicity values at single crystalline zones of elements have shown systematic increase when going from tip to basal cavity. However juvenile elements have lower mosaicity than older one at similar zones. (b) An illustration of crystal domains at different degree of relative mosaicity.

[revised manuscript text omitted]

axis/ c -axis ratio with ontogenetic development observed in this study therefore is likely to have an
impact on preferential uptake of cations during growth. This trend has implications for elemental
concentrations obtained for paleoseawater reconstruction and certainly warrants further
investigation through elemental analysis.

The presence of organic matrix in crystal domains may cause misalignment of mosaic
blocks, thus increasing mosaicity beyond what is observed in purely inorganic apatite minerals
[51]. Moreover, biomineralization processes and the lamellar growth of conodont bioapatite differ
from crystal growth of natural apatite creating differences in crystalline structure. Two clear
patterns of mosaicity are evident in the data presented herein (figure 4) identifying increasing
mosaicity (disordering) of the single crystal domains. Disorder increases both from tip to base in
single specimens, and most importantly, disorder increases with ontogenetic development. The
increased disorder with growth is possibly the result of the growth discontinuities between layer
apposition and the increased potential to have mismatched crystal domains at these discontinuities.
Defects in the alignment at a more juvenile stage will be propagated further with each new layer

added, effectively increasing the degree of disorder with successive stages of development.
Regardless of the cause, the potential significance of this disordering lies in the introduction of
imperfections in either the lattices of individual nanocrystallites themselves or the ordering of
nanocrystallites within each successive layer of crown tissue. Given that elemental uptake and
trapping in biogenic apatite is the result of either adsorption or substitution, the degree of
disordering has the potential to influence the degree to which a conodont may uptake elements
from seawater through either process. Therefore, the variation in disordering (mosaicity) between
crown tissue types, as well as between element types (S, M, etc.), has the potential to influence the
integrity of conodont bioapatite as a paleoceanographic tracer. Care must be taken when
comparing elemental concentrations of different tissue types [14], but based on the data presented
here, it is likely that care must also be taken when comparing chemical concentrations of different
apparatus elements within a single species.

33 290 **5. Conclusions**

Conodont bioapatite is an attractive target as a recorder of ancient seawater chemistry. At present
however, much remains unknown about the controls on elemental uptake, diagenesis, and the
ultimate integrity of conodont bioapatite as a seawater proxy. Whereas clear differences in the
elemental concentrations of different tissue types has already been identified [14], the ultimate
causes and consequences of these differences remain enigmatic. The systematic study of a single
species presented here that utilized a novel approach to both X-ray beam size as well as sample
selection demonstrated variations in crystallography and microstructure due to both ontogeny and
elemental position within the conodont feeding apparatus. These results provide new support for
the importance of functional adaptation in the microstructure of conodonts, as well as new insights

into the possible mechanisms behind the variations in elemental concentration of different crown
tissue types. Ontogeny and elemental position clearly impact crystallographic microstructure and
further systematic geochemical studies that directly consider phylogenetic relationships and
functional morphology are required.

**Data accessibility**

Unit cell values and statistical analysis of the data can be found in the supporting information file.
Diffraction data are deposited in Dryad (<https://doi.org/10.5061/dryad.2rbnzs7jn>)

**Authors' contributions**

310 T.Z.F. and B.D.C. conceived of the study. N.E.B.M. processed the core and recovered the
311 conodont samples utilized. M.S. generated all of the data presented here and drafted the initial
version of the manuscript; all authors contributed to the writing and editing of the final version of
the manuscript.

**Competing interests**

We declare we have no competing interests.

**Funding**

Diffraction studies by M.S. and T.Z.F. was supported by funding from the American Chemical
Society- Petroleum Research Fund (ACS-PRF) Grant # ND-18485100. Conodont extraction and
preparation was supported by United States National Science Foundation Grant # CAREER-
1455030 to B.D.C.

**Acknowledgements**

We thank Dr. Bruce Noll from Bruker AXS for his advice regarding D8 Quest instrument
calibration and Debye ring analysis. We are also grateful to Professor Alejandro Rodriguez-
Navarro (Crystallography and Mineralogy, University of Granada, Spain) who provided access
and advice for using the XRD2DScan software.

**References**

- 1. Epstein AG, Epstein JB, Harris LD. 1977 Conodont color alteration – an index to organic
metamorphism. *USGS Prof. Paper* **995**, 31 pp. (doi:10.3133/pp995)
- 2. Trotter JA, *et al.* 2008 Did cooling oceans trigger Ordovician biodiversification?
Evidence from conodont thermometry. *Science* **321**, 550-554.
(doi:10.1126/science.1155814)
- 3. Trotter JA, *et al.* 2016 New conodont $\delta^{18}\text{O}$ records of Silurian climate change:
Implications for environmental and biological events. *Palaeogeogr. Palaeoclimatol.*
*Palaeoecol.* **443**, 34-48. (doi:10.1016/j.palaeo.2015.11.011)
- 4. Grossman EL, Joachimski MM. 2020 Oxygen Isotope Stratigraphy. In *A Geologic Time*
*Scale 2020* (eds. FM Gradstein, JG Ogg, GM Ogg), in press, Amsterdam: Elsevier.
- 5. Joachimski MM, *et al.* 2009 Devonian climate and reef evolution: insights from oxygen
isotopes in apatite. *Earth Planet. Sci. Lett.* **284**, 599-609. (doi:10.1016/j.epsl.2009.05.028)
- 6. Holmden *et al.* 1996 Isotopic and elemental systematics of Sr and Nd in 454 Ma biogenic
apatites: implications for paleoseawater studies. *Earth Planet. Sci. Lett.* **142**, 425-437.
(doi:10.1016/0012-821X(96)00119-7)

7. Holmden *et al.* 1998 Isotopic evidence for geochemical decoupling between ancient
epeiric seas and bordering oceans: Implications for secular curves. *Geology* **26**, 567-570.
(doi:10.1130/0091-7613(1998)026<0567:IEFGDB>2.3.CO;2)
8. Trotter JA, *et al.* 2015 Long-term cycles of Triassic climate change: a new $\delta^{18}\text{O}$ record
from conodont apatite. *Earth Planet. Sci. Lett.* **415**, 165-174.
(doi:10.1016/j.epsl.2015.01.038)
9. Wright J, Seymour RS, Shaw HF. 1984 REE and Nd isotopes in conodont apatite:
Variations with geological age and depositional environment. *GSA Spec. Pap.* **196**, 325-
340.
10. Zhang L, *et al.* 2016 Diagenetic uptake of rare earth elements by conodont apatite.
*Palaeogeog. Palaeoclimatol. Palaeoecol.* **458**, 176-197.
(doi:10.1016/j.palaep.2015.10.049)
11. Zhao L, *et al.* 2013 Rare-earth element patterns in conodont albid crowns: Evidence for
massive inputs of volcanic ash during the latest Permian biocrisis? *Glob. Planet. Change*
**105**, 135-151. (doi:10.1016/j.gloplacha.2012.09.001)
12. Chen J, *et al.* 2015 Diagenetic uptake of rare earth elements by bioapatite, with an
example from Lower Triassic conodonts of South China. *Earth-Science Rev.* **149**, 181-
202. (doi:10.1016/j.earscirev.2015.01.013)
13. Felitsyn S, *et al.* 1998 Nd isotope composition and rare earth element distribution in early
Paleozoic biogenic apatite from Baltoscandia: A signature of Iapetus ocean water.
*Geology* **26**, 1083-1086. (doi:10.1130/0091-7613(1998)026<1083:NICARE>2.3.CO;2)
14. Trotter JA, Eggins SM. 2006 Chemical systematics of conodont apatite determined by
laser ablation ICPMS. *Chem. Geol.* **233**, 196-216. (doi:10.1016/j.chemgeo.2006.03.004)

15. Cummins RC, *et al.* 2014 Carbonate clumped isotope constraints on Silurian ocean
temperature and seawater $\delta^{18}\text{O}$. *Geochim. Cosmochim. Acta* **140**, 241-258.
(doi:10.1016/j.gca.2014.05.024)
16. Wenzel B, Lécuyer C, Joachimski MM. 2000 Comparing oxygen isotope records of
Silurian calcite and phosphate – $\delta^{18}\text{O}$ compositions of brachiopods and conodonts.
*Geochim. Cosmochim. Acta* **64**, 1859-1872. (doi:10.1016/S0016-7037(00)00337-9)
17. Pander CH. 1856 *Monographie der fossilen Fische des silurischen Systems der*
*russischbaltischen Gouvernements*. St. Petersburg: Akademie der Wissenschaften.
18. Stauffer CR, Plummer HJ. 1932 Texas Pennsylvanian conodonts and their stratigraphic
relations. *Univ. Texas Bull.* **3201**, 13-50.
19. Murdock DJE, *et al.* 2013 The origin of conodonts and of vertebrate mineralized
skeletons. *Nature* **502**, 546-549. (doi:10.1038/nature12645)
20. Donoghue PCJ. 1998 Growth and patterning in the conodont skeleton. *Phil. Trans. R.*
*Soc. Lond. B* **353**, 633-666. (doi:10.1098/rstb.1998.0231)
21. Donoghue PCJ. 2001 Microstructural variation in conodont enamel is a functional
adaptation. *Proc. R. Soc. Lond. B* **268**, 1691-1698. (doi:10.1098/rspb.2001.1728)
22. Trotter JA, *et al.* 2007 New insights into the ultrastructure, permeability, and integrity of
conodont apatite determined by transmission electron microscopy. *Lethaia* **40**, 97-110.
(doi:10.1111/j.1502-3931.2007.00024.x)
23. Branson EB, Mehl MG. 1933 Conodonts from the Bainbridge (Silurian) of Missouri. In
*Conodont Studies Number 1*. Univ. of Missouri Studies, A Quarterly of Research **8**, 39-
53.

24. McAdams NEB, *et al.* 2019 Integrated $\delta^{13}\text{C}_{\text{carb}}$, conodont, and graptolite
biochemostratigraphy of the Silurian from the Illinois Basin and stratigraphic revision of
the Bainbridge Group. *GSA Bull.* **131**, 335-352. (doi:10.1130/B32033.1)
25. Melchin MJ, Sadler PM, Cramer BD. 2020 The Silurian Period. In *A Geologic Time*
*Scale 2020* (eds. FM Gradstein, JG Ogg, GM Ogg), in press, Amsterdam: Elsevier.
26. Jeppsson L, Anehus R. 1995 A buffered formic acid technique for conodont extraction. *J.*
*Paleont.* **69**, 790-794. (doi:10.1017/S0022336000035319)
27. Cooper BJ. 1976 Multielement conodonts from the St. Clair Limestone (Silurian) of
southern Illinois. *J. Paleont.* **50**, 205-217.
28. Barrick JE. 1977 Multielement simple-cone conodonts from the Clarita Formation
(Silurian), Arbuckle Mountains, Oklahoma. *Geologica et Palaeontologica* **11**, 47-68.
29. Klapper G, Philip GM. 1971 Devonian conodont apparatuses and their vicarious skeletal
elements. *Lethaia* **4**, 429-452. (doi:10.1111/j.1502-3931.1971.tb01865.x)
30. Novak GA, Colville AA. 1989 A practical interactive least-squares cell-parameter
program using an electronic spreadsheet and a personal computer*. *Amer. Mineral.* **74**,
488-490.
31. He BB. 2018 Data Treatment. In *Two-dimensional X-ray Diffraction*, pp. 151-190. 2nd
Edn. Hoboken, NJ: John Wiley & Sons.
32. Kniep R, Busch S. 1996 Biomimetic growth and self-assembly of fluorapatite aggregates
by diffusion into denatured collagen matrices. *Angewandte Chemie Internat. Ed. in*
*English* **35**, 2624-2626. (doi:10.1002/anie.199626241)

33. Rodriguez-Navarro A. 2006 XRD2DScan: new software for polycrystalline materials
characterization using two-dimensional X-ray diffraction. *J. App. Crystall.* **39**, 905-909.
(doi:10.1107/S0021889806042488)
34. Giacovazzo C, *et al.* 2011 *Fundamentals of crystallography*. Oxford: Oxford University
Press.
35. Dobrianov I, *et al.* 1999 X-ray diffraction studies of protein crystal disorder. *J. Cryst.*
*Growth* **196**, 511-523. (doi:10.1016/S0022-0248(98)00833-1)
36. Sauter NK, Zwart PH. 2009 Autoindexing the diffraction patters from crystals with a
pseudotransition. *Acta Crystall. Sect. D* **65**, 553-559. (doi:10.1107/S0907444909010725)
37. Tadano S, Giri B. 2012 X-ray diffraction as a promising tool to characterize bone
nanocomposites. *Sci. Tech. Adv. Mater.* **12**, 064708. (doi:10.1088/1468-
6996/12/6/064708)
38. Li Z, *et al.* 2015 A mineralogical study in contrasts: highly mineralized whale rostrum
and human enamel. *Sci. Rep.* **5**, 16511. (doi:10.1038/srep16511)
39. Medici L, *et al.* in press Mineralogy and crystallization patterns in conodont bioapatite
from first occurrence (Cambrian) to extinction (end-Triassic). *Palaeogeog. Palaeoclimat.*
*Palaeoecol.* (doi:10.1016/j.palaeo.2019.02.024)
40. Hughes JM, Cameron M, Crowley KD. 1989 Structural variations in natural F, OH, and
Cl apatites. *Amer. Mineral.* **74**, 870-876.
41. Hughes JM, Rakovan J. 2002 The crystal structure of apatite, $\text{Ca}_5(\text{PO}_4)_3(\text{F},\text{OH},\text{Cl})$. *Rev.*
*Mineral. Geochem.* **48**, 1-12. (doi:10.2138/rmg.2002.48.1)
42. Reynard B, Lécuyer C, Grandjean P. 1999 Crystal-chemical controls on rare-earth
element concentrations in fossil biogenic apatites and implications for

paleoenvironmental reconstructions. *Chem. Geol.* **155**, 233-241. (doi:10.1016/S0009-
2541(98)00169-7)
43. Frank-Kamenetskaya OV, *et al.* 2014. Refinement of apatite atomic structure of albid
tissue of Late Devon conodont. *Crystall. Rep.* **59**, 41-47.
(doi:10.1134/s1063774514010039)
44. Madupalli H, Pavan B, Tecklenburg MMJ. 2017 Carbonate substitution in the mineral
component of bone: Discriminating the structural changes, simultaneously imposed by
carbonate in A and B sites of apatite. *J. Sol. State Chem.* **255**, 27-35.
(doi:10.1016/j.jssc.2017.07.025)
45. Hovis G, *et al.* 2015 Thermal expansion of F-Cl apatite crystalline solutions. *Amer.*
*Mineral.* **100**, 1040-1046. (doi:10.2138/am-2015-5176)
46. Chen J, *et al.* 2015 Diagenetic uptake of rare earth elements by bioapatite, with an
example from Lower Triassic conodonts of South China. *Earth-Sci. Rev.* **149**, 181-202.
(doi:10.1016/j.earscirev.2015.01.013)
47. Zhang L, *et al.* 2017 Raman spectral, elemental, crystallinity, and oxygen-isotope
variations in conodont apatite during diagenesis. *Geochim. Cosmochim. Acta* **210**, 184-
207. (doi:10.1016/j.gca.2017.04.036)
48. Cacciotti I. 2016 Cationic and anionic substitutions in hydroxyapatite. In *Handbook of*
*Bioceramics and Biocomposites* (ed. IV Antoniac), pp. 145-211. Switzerland: Springer
International Publishing.
49. El Feki H, Savariault JM, Ben Salah A. 1999 Structure refinements by the Rietveld
method of partially substituted hydroxyapatite: $\text{Ca}_9\text{Na}_{0.5}(\text{PO}_4)_{4.5}(\text{CO}_3)_{1.5}(\text{OH})_2$. *J. Alloys*
*Compounds* **287**, 114-120. (doi:10.1016/S0925-8388(99)00070-5)

50. El Feki H, *et al.* 2000 Sodium and carbonate distribution in substituted calcium
hydroxyapatite. *Sol. St. Sci.* **2**, 577-586. (doi:10.1016/S1293-2558(00)01059-1)
51. Vallet-Regi M, Arcos Navarrete D. 2016 Biological apatites in bone and teeth. In
*Nanoceramics in clinical use: From materials to applications*, Ed. 2, pp. 1-29.
Cambridge: The Royal Society of Chemistry. (doi:10.1039/9781782622550)
52. Ferretti A, *et al.* 2017 Diagenesis does not invent anything new: Precise replication of
conodont structures by secondary apatite. *Sci. Rep.* **7**, 1624. (doi:10.1038/s41598-017-
01694-4)
53. Shirley B, *et al.* 2018 Wear, tear and systematic repair: testing models of growth
dynamics in conodonts with high-resolution imaging. *Proc. R. Soc. B.* **285**, 20181614.
(doi:10.1098/rspb.2018.1614)

Appendix B

COLLEGE OF
LIBERAL ARTS & SCIENCES

Department of Chemistry

E331 Chemistry Building
Iowa City, Iowa 52242-1294
319-335-1350 Fax 319-335-1270

June 6, 2020

To:
Professor Rachel Wood
Associate Editor, *Royal Society Open Science*

Sub: Revision requested for RSOS-200322

Dear Prof. Rachel Wood

Please consider our revised manuscript “**Ontogenetic variability in crystallography and mosaicity of conodont apatite: Implications for microstructure, paleothermometry and geochemistry**”. We thank the reviewers for their helpful suggestions and have tried to address their concerns below and in the revised manuscript. The reviewer’s comments appear in italics and our responses occur after the comments.

Reviewer-1

Page: 4

Line 38

While this is true, it rather underplays some of what is already known. For example, Trotter and Eggins 2006 (already cited elsewhere in this ms) noted differences in susceptibility of different conodonts hard tissues to post mortem chemical alteration and demonstrated that white matter is more resistant. previous work should be properly acknowledged, and novelty of current work should not be overstated (there is novelty; putting it in context does not diminish this). Similarly previous, albeit preliminary work using EBSD to investigate conodont crystallinity is not mentioned in this ms (Perez-Huerta et al Lethaia 2012)

Thank you for your suggestion regarding additional references to include in our manuscript. We have added some additional discussion about finding by Trotter and Eggins 2006, Perez-Huerta et al Lethaia 2012 in page 2 line 41 (Reference no 14 and 17).

Line 43

standardizing terminology at this point in the ms would help the reader, esp if it aligns with general usage. The modern paleobiological/phylogenetic/histological conodont literature generally uses "lamellar crown tissue" and 'white matter'. 'albid' and 'hyaline' are generally redundant and a little archaic. (I also note that in the glossary of the conodont treatise -

admittedly a little old now - albid and hyaline refer to whole elements, not tissues. I suggest 'lamellar crown tissue' and 'white matter' throughout

We have made the requested changes throughout the manuscript.

Line 45

Inserted Text composed

Word “comprised” changed into “composed” accordingly.

Page: 5

Line 49

Donoghue and Purnell 1995 Geology worth citing here; Shirley et al essentially corroborated the conclusions of this work.

Thank you for suggesting additional literature for us to consider. We could not find the work by Donoghue and Purnell in *Geology* for the year 1995, but we did find work by these authors in an article published in 1999. We believe this is the reference the reviewer is referring to because this is the only work by Donoghue and Purnell published in *Geology*. We have included this citation in our manuscript as suggested in page-3, line-52 (reference no: 27).

Line 50

This seems a rather harsh characterisation which implies previous studies were poorly designed. The reality in most cases is that they were designed to answer a question that is different to that upon which this ms focusses.

We did not mean to imply the previous studies were poorly designed, just that they chose to analyze the data in a manner that resulted in no apparent trends by aggregating all data together. It is important to point out that aggregation of the data led to the conclusion that there is no relationships between ontogeny, species, geologic age, or element and diffraction features. It is incorrect to state that they were answering different questions. They were attempting to relate unit cell parameters to different element features, but did not consider species, type, or ontogeny. This limitation sets up the major focus of our study.

We have altered the language slightly to address the concerns of the reviewer. The original language was:

“Whereas the significance of phylogeny and ontogeny in microstructure has been well documented [21,53,55], nearly all modern studies of conodont crystallography have chosen to combine crystallographic data from different species, different element types, different intervals of Earth history, and different stages of ontogenetic development [39,47,52,58]. The net result has often been aggregated data that show no significant trends and a conclusion that conodont bioapatite crystallography is essentially identical across the clade.”

And we have altered it to:

“Whereas the significance of phylogeny and ontogeny in microstructure has been well documented [21,53,55], many studies of conodont crystallography combine crystallographic data from different species, different element types, different intervals of Earth history, and different stages of ontogenetic development [39,47,52,58]. The net result is aggregated data that do not display significant trends and a conclusion that conodont bioapatite crystallography is essentially identical across the clade.”

Page: 6

Line 76

*I find this problematic. Because no articulated skeletons of *Dapsilodus* are known, there is no evidence that allows determination of what size elements occurred together in an apparatus. Furthermore, as noted below, we don't actually know how many of each element morphotype occurred in the apparatus, and whether duplicate elements of similar morphotype were of similar size. I do not think, given these uncertainties, that the authors can meaningfully talk about ontogenetic variation. They are talking about variation with element size and robustness. A reasonable case could be made that, for a particular element morphotype, this is likely to reflect ontogeny to some degree. However, different element types cannot be compared, and the Sb-c category could hide elements from multiple different locations. In the absence of fossils preserving an articulated apparatus, these uncertainties cannot be addressed, and I suggest the authors recast all aspects of this ms relating to 'ontogeny' in this light. I don't think elements of *dapsilodus* can meaningfully be classified as 'juvenile' and 'older'.*

*Wear is a red herring here, because from what we know of conodont development through life (noting that *dapsilodus* hasn't been studied, as far as I know). Multiple cycles of growth, use and wear, mean that wear can occur in elements of different sizes and ages (see Donoghue and Purnell 1999, *Geology*)*

We respectfully disagree with the reviewer on several grounds and have added new plates to help demonstrate the point. 1) If they were referring to “natural assemblages” of the entire feeding apparatus of a given species, there are likely fewer than 250 known worldwide. If they are referring to soft-body impressions (as conodonts contained no skeleton), the number is likely less than 50 ever recovered. The collection of *Dapsilodus* from the Schlamer core contains more than 1,000 elements of *Dapsilodus*, and illustrates clear ontogenetic growth series of each element type. We have included a new figure (figure 1) that illustrates representative growth series for S and M elements as well as an improved version of figure 2. We have hundreds of each element that allow a clear picture of growth within the species. We dramatically improved the figures of each element with clearer synoptic presentation so that the clear size differentiation of each element in each category can be much better identified by the reader. We suggest the changes illustrated in the new figure 1, combined with our grouping in figure 2, are most parsimoniously explained via ontogeny.

Line 79

This is a pretty poor quality figure. As a minimum, given that one of its primary purposes is to demonstrate size variation between classes, all elements should be shown at the same

magnification. Aesthetic consideration are more subjective, but to my eye this is poorly put together and scruffy looking.

We have provided a new version of this original figure in synoptic presentation as requested. All specimens are now shown in equal magnification as requested. We thank the reviewer for pointing this out and the revised figure, combined with the new figure 1 should make this much clearer.

Page: 7

Line 87

Counts and ratios of different element morphotypes in collections of isolated elements are known to be a very unreliable guide to the numbers of elements in an articulated apparatus. As far as I know, in all cases where hypothetical ratios of isolated element types have been compared to the reality of actual articulated apparatuses, the hypothetical ratios have been shown to be incorrect. Although not explicitly addressing ratios in coniform conodonts, Purnell and Donoghue 2005 (Special papers in Palaeontology) discuss the biases.

Whereas there have been some illustrations of ratio counts being incorrect, it is not true that all cases have been shown to be incorrect. Furthermore, with due respect, we do not believe that it is relevant to the current discussion. In the end it does not matter if we have the complete apparatus of *D. obliquicostatus*, nor does it matter if we even have the elements in their correct position. The point being that they demonstrate different element types and that there is systematically different crystallography between the types. Furthermore, the identification of Sa elements as Sa is quite independent of the argument made by the reviewer. Not only are they verified as Sa by the ratio counts, but their symmetry is the primary reason why they are regarded as Sa. Morphology is what identifies the M elements in this case as M, not ratio counts. The Sb, Sc, vs. Sb-c was done specifically by Barrick to address the problem highlighted by the reviewer. Although the reviewer is correct that we do not know with certainty if these were a single position or two (or more positions), it is once again not entirely germane to the discussion. The major and most important points are that all elements utilized here belong to the same species and this has been done following the procedures of Klapper and Philip (1971), and that there are differences between element positions. Clearly M and Sa can be identified as their respective groups, and the Sb-c (or Sb or Sc) can be identified as a different position or positional group. The point still stands that they are a different elemental position and that this is being reflected in our data.

Line 93

I find this a little confusing. is a sample a spot on an element or an element? 'sample' and 'element' seem to be used almost, but not quite, interchangeably.

Yes, by “a sample” we also meant “an element”.

More importantly, these zones are later considered comparable (i.e. zones 1-3 in 'juvenile' elements are compared with zone 1-3 in 'older' elements. Leaving aside the question of whether these categories reflect ontogeny, these zones are not comparable in this way because of the way conodont elements add lamellae. They do not grow by basal addition, but by appositional growth, so the tip of a smaller/'juvenile' element is not homologous with the tip of a larger/'older' element, because the earlier growth stage is subsumed within the later one. The only comparable point - the only landmark that allows 'zones' in one element to be compared to those in another, is the tip of the basal cavity. otherwise comparisons between elements of different sizes are meaningless. The least comparable parts of elements of different sizes are the tips.

The argument that the tip of a juvenile is not “homologous” with the tip of an adult is confusing and, with due respect, we do not fully agree with the reviewer. To start with, homology is about the shared ancestry of a feature between two or more species. Typically, the only time homology is used within a single species would be in the context of serial homology...as in my thumb is the serial homolog of my big toe... So the reviewer’s comments are suggesting that the tip of juvenile has a different ancestry than the tip of an adult...of the same species and same element type. That would be like saying the tip of my thumb as an adult has a different ancestry than the tip of my thumb when I was younger.

Of course conodonts grow by apposition, and yes of course the tip of a juvenile is subsumed within later growth, but those points do not at all lead to the conclusion they presented. The crystallography of the tip created during earlier (or juvenile) stages of growth are exhibiting a different crystallography than the tip created during later stages. Yes, of course the later tip grew later by apposition, and that is kind of the point we are making here...It is not that the “juvenile tip” somehow morphs into a different crystallography. Another way to think about this is what a researcher would be choosing as a point to, for example, shoot with LA-ICPMS for chemical analysis. If they chose the tip of a juvenile, would that be identical crystallography to the same elemental position (i.e. tip) of the same species, same element, of an adult? These differences are precisely the point and the reviewer’s argument actually supports this point.

*Line 94
implies that not all analysis were conducted in the same way*

We can assure the reviewer that all analyses were conducted carefully following the same standard procedure. We have changed the sentence into “In each experiment, conodont elements were coated with mineral oil, then attached on top of MiTeGen Dual-Thickness MicroLoop in a vertical position.” (Page 5, Line 106)

*Page: 8
Line 106
these seem to be critical to results and interpretation, but are not explained in a way that someone whose expertise is in conodonts would understand. Presumably, this is the target audience, so an explanation would be worthwhile.*

This is a valid point and we thank reviewer for that. We have added a brief explanation regarding X-ray diffraction and related data analysis procedures at the beginning of the results section to give context to our data (page 7, line 147):

“Crystalline materials interact with X-ray radiation to give a characteristic diffraction pattern due to coherent scattering of these electromagnetic waves. Materials typically lie within a spectrum of crystallinity that ranges from the ideal single crystalline to polycrystalline and finally amorphous, which are defined by the relative length of the coherent scattering domain, the relative disorder of those domains, and their overall orientation [44, 45]. Materials composed of highly ordered atomic lattices (e.g. single-crystals) display diffraction individual reflections (observed as spots on the 2D area detectors) that are indicative of the large domains that diffract with coherent scattering [46]. Polycrystalline materials are composed of small (5-10 μm) crystallites that are arranged in random orientations throughout the sample [44]. Coherent scattering of the X-rays in a polycrystalline sample results in a series of diffraction rings (Debye rings). Amorphous materials may have large particle sizes, but the coherent scattering domains are small (<5 nm) and result in broad, low-intensity features in the 2-D images [44-46]. Disorder in coherent domains of single crystalline structures will change the angle of diffracted X-rays corresponding to related crystallographic planes in varying degrees; thus will change the shape and width of diffraction features [44, 47]. In the case of polycrystalline materials, if the small single crystallites are not randomly oriented then some of the crystallographic planes will interact with incoming X-rays more often, thus diffracted X-rays from those planes will create more intense regions in the diffraction pattern [48]. Variations in the intensity of the Debye rings in diffraction pattern is a phenomenon known as texturing [48]. As X-rays interact differently depending on crystalline structure, we determined unit cell parameters of conodont zones using separate protocols suitable for either single crystalline or polycrystalline material.”

In addition, we have also added a scheme that depicts the details of the X-ray diffraction experiment in electronic supplementary info (Scheme S1). We hope this image will also clarify the experimental X-ray diffraction procedure.

Line 115

methods do not discuss statistics and statistical methods. e.g. normality of distributions, appropriateness of parametric tests, methods for regression fitting, approaches to significance testing (alpha values etc).

We have significantly improved the discussion of the statistics as well as the statistical treatment overall and thank the reviewer for pointing this out. We have included the text below, as well as a wealth of new information in the Electronic Supplement. The following paragraph in the methods section address the reviewer’s comments.

“Quantile-quantile (qq) plots of single crystalline zones are provided in Figure (S2) as a test of normal distribution of unit cell parameters. Given that neither the measurements on a nor c are truly “controlled” and thus each independent variables, model II linear regressions (Ordinary Least Squares [OLS], Major Axis [MA], Standard Major Axis [SMA], and Ranged Major Axis [RMA]) were carried out to determine the association of the unit cell parameters. A model II OLS regression is the bifurcation of the “x on y” and “y on x” regressions. Parametric p-values for OLS and permutational probability tests (for MA and RMA) were calculated (n=10,000) to determine whether the correlations were statistically significant. Regression analysis was performed in ‘R’ utilizing the “lmodel2” package [42]. Following the decision tree in Legendre and Legendre [43] and Legendre [42], the random variation (i.e., error variance) on both variables (a and c) are approximately equal, the data have an approximately bivariate normal distribution (figure S2), and

both variables are expressed in the same physical units, therefore major axis regression (MA) should be utilized and is shown in figure 4. All regression types are shown in the supplementary file (figures S3,S4, tables S6,S7) and major-axis (MA) regressions are shown below.”

Line 119

I found it difficult to see the patterns that the authors describe. Non-expert readers not used to interpreting this type of data will need a more detailed description if they are to follow how the patterns in the data support the interpretations presented

We appreciate this feedback. We have combined this issue with the reviewer’s earlier suggestion to provide additional background on X-ray diffraction and was again addressed on page 7, line 147. We have also improved the contrast and quality of images showing XRD diffraction pattern (Revised Figure 4).

Line 119

I struggled to see the distinct differences highlighted by the authors below; without this its hard to judge the degree to which the data support the interpretations, and this should not be a matter of trust.

We understand that to those outside the field of crystallography it may be difficult to interpret the data on the images alone. We have included improved images that enhance the contrast and readability of each of the XRD patterns. We have also deposited the diffraction images in suitable format at Dryad data repository (<https://doi.org/10.5061/dryad.2rbnzs7jn>) as per recommendation of Royal Society- Open Science, so that readers who are experts in interpreting 2D X-ray diffraction images can have access to the raw images. We then provided additional analysis (e.g. mosaicity calculation, psi scan and unit cell parameter) of these images to be able to relate to variability of crystalline structure along the length and in between older and juvenile conodont elements. This provides additional details and insights to those readers in the conodont community.

Line 119

Presumably, as whole elements were analysed, the X-ray beam passes through surface layers of lamellar crown tissue before it reaches the white matter. If so, do the patterns from white matter zones reflect a signal that mixes white matter and lamellar crown?

The reviewer is correct in stating that our X-ray diffraction method is not a surface technique. In the case of single crystal X-ray diffraction, we operate in transmission mode, where the X-ray beam penetrates the whole sample and we collect the diffracted beam related to the zone where the X-ray passed through. This gives us an understanding of the structural features within that region. The major significance of this technique is that it allows us to identify the two tissue types and to separate them in the analysis. The software easily identifies the presence of two distinct domains of coherent diffraction (polycrystalline and single crystalline) and we can accurately differentiate based upon integration of diffraction spots versus rings.

Page 11

Fig 2 caption:

I have no experience of interpreting plots like this, but the images themselves seem to lack contrast, which makes it hard to pick out patterns clearly.

Could contrast be enhanced, or is absolute intensity part of the signal?

We have improved each of the images in photoshop to increase the contrast and readability of diffracted X-rays as requested. We have also deposited all of our X-ray diffraction images in an open access database Dryad (<https://doi.org/10.5061/dryad.2rbnzs7jn>) as per recommendation of Royal Society- Open Science.

Page 12

Line 137

This seems to be methods

That particular discussion have been moved into method section according to your suggestion.

Page: 14

Figure 3 caption

the combining of data in these plots and the statistical tests reported as if they represent categories that are directly comparable is wrong, for the reasons outlined above. 'Older' elements of different morphotypes (M, Sa, and Sb-c) cannot meaningfully be combined. Same applies to 'juvenile'.

As stated above, with due respect, we do not agree with the reviewer on two major points. 1) we believe that we can easily demonstrate ontogeny in the samples, as demonstrated in the new figures, and 2) whereas this was not an original objective of this study, but when the data plotted as what appear to be separated clouds, based upon juvenile and older populations, we believed it is necessary to point this out, both graphically and in text. As discussed in the revised text, there is still much work to be done regarding statistical significance by improving the total sample size of similar studies.

Figure 3 caption

Also, the plots show what I assume are regressions. these are problematic. Firstly, there is no indication of how well they represent the data. more importantly, the text suggests that there is no reason to chose a or c as dependent or independent variable, and both are equally prone to measurement error. in which case simple least squares linear regression is not appropriate. reduced major axis or orthogonal regression should be used.

We thank the reviewer for this comment and we have significantly improved the statistical treatment of the unit cell data presented. There is new discussion, plots and details about the statistical analysis in our revised manuscript.

Page: 14

Figure 3 caption

no P value reported

We have significantly improved the treatment of statistics throughout the manuscript and new p values have also been reported.

Page: 15

Line 161-164

this mode of reporting looks odd; is it intended to imply an alpha value of 0.05 as a threshold for significance?

We removed the sentence in question and have significantly improved our statistical treatment as discussed above.

Line 165

no P values

We removed the sentence in question and have significantly improved our statistical treatment as discussed above.

Page: 16

Fig 4 caption

as noted above, zones are not comparable in this way; tips are not homologous in 'older' and 'juvenile' elements.

The term “tip” is used as a general term to mark where the diffraction measurement was started along the length of each element. We started our diffraction experiment from tip and went along the length of the elements to basal cavity. As stated above, with due respect, the argument about homology presented by the reviewer is contentious and we do not believe to be applicable.

Page: 16

Fig 4 caption

Regarding b, I'm not sure that the diagram is an adequate representation of increasing mosaicity as it implies disruption rather than variation in crystallite orientation

The mosaicity describes the degree of crystalline imperfections and describes a crystal as composed of mosaic blocks that can have different levels of alignment. The misalignment or misorientation can happen in any direction of crystallite block. The X-ray diffraction measures an average value of mosaicity from all crystallographic plane in degree. The figure that we constructed represented misalignment at any possible direction. In fact, that's how they were represented in other peer-reviewed publications (e.g. Snell, E. H., Helliwell, J. R. 2005 Macromolecular crystallization in microgravity. Reports on Progress in Physics. 68, 799-853. Ruyters, G., Betzel, C. 2017 Protein Crystallization in Space: Early Successes and Drawbacks in the German Space Life Sciences Program. In Biotechnology in Space., pp. 11-26. Cham: Springer International Publishing.) However, we have realized that using size of different crystal domains to draw the figure may confuse reader as we are taking about same single crystalline materials. So we changed the figure accordingly. We also added the following statement to the

caption: “The orientation of the crystalline domains is exaggerated in this image to illustrate the idea of mosaicity.”

Line 192

an expanded version of this section, earlier in the ms, would help the reader to understand the results

In response to your earlier suggestion, we have added a brief description about relationship between materials of different crystalline structure and their X-ray diffraction pattern (page 7, line 147). We have included additional discussion about texturing in that new section.

Page: 18

Line 230

more detailed subsequent work perhaps also worth citing - e.g. Martinez-Perez et al Palaeontology 2014.

The citation have been included according to your suggestion. Page 13, line 254, citation no: 54.

Page: 20

Line 264

somewhere, whether in the introduction or here, the authors need to acknowledge that the variation in delta 18 O within conodonts and between elements in the same taxon, has been investigated directly. Wheeley et al 2012 (J Geol Soc London) found minimal difference between white matter and lamellar crown tissue. It thus appears that the conclusion drawn here, that crystallographic differences are likely to have an impact of geochemical signals in conodont elements, is unsupported by available data.

We respectfully disagree with this point. To begin with, the issue of $\delta^{18}\text{O}$ having any bearing on elemental abundance within the conodont does not make sense. If $\delta^{18}\text{O}$ was being impacted, it would be the result of fractionation during crystallization and incorporation into the PO_4 . Hence that would be an issue of crystallographic fractionation of isotopes, which is a completely different issue than elemental abundance data from conodonts. The points we are making here, similar to the work of Trotter and Eggins, etc., were about elemental abundance. Aside from the significant complications of Wheeley et al. (2012), there are multiple data sets available that demonstrate there are significant species-dependent vital effects on conodont $\delta^{18}\text{O}$ (e.g., the forthcoming chapter on $\delta^{18}\text{O}$ stratigraphy in the GTS2020 by Grossman and Joachimski)...one likely cause of which is crystallographic differences between species... We chose not to promulgate the conclusions of Wheeley et al (2012) specifically because they are not supported by further data from multiple species.

Line 276

this is not how conodont elements grow. they grow through distinct phases, with prolonged period of function between. defects are not propagated between phases. See Donoghue and

Purnell 1999 Geology, and Shirley et al 2018 (already cited)

Yes, of course they grow in distinct phases. However, that does not somehow mean that defects cannot be propagated between phases, nor has this ever been conclusively demonstrated. Our data suggest an increase in disorder during growth. We supplied a potential cause as a hypothesis and stated that “The increased disorder in growth is possibly the result of growth discontinuities between layer apposition and the increase potential to have mismatched crystal domains at these discontinuities.” Firstly, we clearly stated exactly what the reviewer did...that conodonts grow in layers in distinct phases. We changed the wording of the following sentence to read “Defects in the alignment at a more juvenile stage *could* (was will) be propagated further with each new layer added...”. We are attempting to explain our data by presenting a hypothesis that is not in violation of anything in the literature. There is not a clear demonstration that defects are not propagated between phases and allows us to postulate hypothesis for further study.

Page 21

Line 287

Addressed by Wheeley et al 2012 J Geol Soc London

With due respect, we have to state that this was not addressed by Wheeley et al 2012. That manuscript is about oxygen isotopic composition. These are very different things controlled by different chemical and physical processes.

Reviewer-2

METHODS

The method employed in the study allows in situ X-ray diffraction and deriving unit cell parameters and subsequent crystallographic parameters such as mosaicity from the diffraction of an X-ray beam on the surface of the specimen. Diffraction and effective analysed area is affected by sample topography and quantitative unit cell analyses should be performed on flat, polished surfaces to minimise diffraction artifacts. Conodont element surface has curvature that is substantial compared to the spot size used (120 μm).

We think the reviewer is confusing XRD techniques and is confusing reflection mode with transmission mode. The transmission mode, utilized herein, is specifically designed to NOT be influenced by topography. In the current study, we have used a suitable strategy, instrument and set up for X-ray diffraction experiment to collect high resolution data and minimize diffraction artifacts. The reviewer is incorrect that quantitative unit cell analysis should be performed on a flat polished surface. If this was the case, then unit cell determination on mineralogical specimens could not be quantitative and this is clearly untrue. Utilizing single-crystal diffraction provides quantitative unit cells with errors of typically (0.0001 Å) for crystalline materials and usually (0.001 Å) for polycrystalline materials. The curvature does change the potential area of collection, but that just changes the intensity of the diffraction spots, not the position. It is the position of the diffraction features that determines the unit cell parameters.

Moreover, the study makes a point about crystallographic parameters changing along the element with lower mosaicity towards the cusp. This ignores the effect of histology: the measurement effectively compares two types of tissues which are known already to consist of crystals with largely differing sizes and ordering. The cusp consists of the albid tissue which, going down the element, is increasingly embedded in hyaline tissue. The difference in mosaicity presented in the study is not meaningful if these systematic differences of tissues are not accounted for. This is virtually impossible when measuring element surface, which will provide a mixture of values for both tissues, and would require a sectioned specimen.

Perhaps the reviewer is more familiar with electron microscopy or even powder X-ray diffraction, where you are working in reflection mode. Single-crystal X-ray diffraction works in transmission mode, so we are penetrating the entire sample. This enables use to collect information on the albid and hyaline tissue simultaneously. So, once again as discussed above with Reviewer #1, we are able to identify both tissue types simultaneously and this is the reason why this comment is not valid, nor is the previous comment valid about the need for a flat surface.

Previous studies indicated that the albid region behave more like single crystal with coarse size crystallites arranged in regular ordering, in contrast basal body contains fine, disarrayed randomly oriented crystallites (Trotter, J. A., Gerald, J. D. F., Kokkonen, H., Barnes, C. R. 2007 New insights into the ultrastructure, permeability, and integrity of conodont apatite determined by transmission electron microscopy. *Lethaia*. 40, 97-110 and Donoghue, P. C. J. 1998 Growth and patterning in the conodont skeleton. *Philos Trans R Soc Lond B Biol Sci*. 353, 633-666). The presence of two structurally distinct tissue type demands a special strategy for analyzing their crystalline structure. To tackle this challenge, we have used 2D X-ray diffraction technique that is capable of recording X-ray diffraction pattern of both single crystalline and polycrystalline material. The Bruker Inc. provided Apex-3 software system further can calculate different crystallographic parameter separately for single crystalline and polycrystalline region. In addition, we have also used more sophisticated analysis through other software (e.g. XRD2D scan for texturing). Separation of the single-crystal diffraction spots associated with the albid region allows us to quantitatively calculate the mosaicity for that specific tissue type.

This is the most likely explanation for lower polycrystallinity of M elements compared to others: this might be simply due to a higher proportion of albid tissue.

This would be true if we were unable to separate the signal from each tissue type when they were analyzed together, but as we explained earlier, this is not the case. The methodology allows us to explicitly evaluate both tissue types simultaneously.

The same limitation applies to the proposed observation of increasing a-axis/c-axis ratio during growth: the authors attribute this to a change in chemical composition, but do not compare it with actual data, which is available (e.g. Trotter et al. 2007), and do not consider differences between tissues, which are the most likely explanation.

We are confused by this argument. A polycrystalline sample and a single crystal of the same mineral with the same composition will have the same unit cell parameters. These unit cell parameters will change if there are compositional changes, such as described by Vegard's Law.

This is basic mineralogy. And furthermore, we are comparing it to actual data...i.e. this manuscript. The basic premise of the method employed here is to specifically be able to determine precisely this. We cannot possibly compare it to anything more “actual” than the data we collected.

The authors consistently refer to areas with low mosaicity as “single crystal” or “single crystalline”, which is very confusing, since they themselves document that they consist of multiple nanocrystals. A more appropriate term would be needed.

We have added additional descriptions regarding the crystallinity and relationship to X-ray diffraction (as described in the response to reviewer 1). That new discussion (Page-7, Line-147-167) explained the basis of terms have been used in current manuscript.

Preferred crystallographic orientations are not meaningful if they are not provided with respect to sample orientation (lines 192-206).

Thank you for this suggestion. We have added a new figure in the supplementary file (Figure S1) about X-ray diffraction experiment. The Scheme clearly demonstrate how sample was oriented during data collection. In addition to that, the goniometer orientation (2θ , ω , Φ , and χ) is also in the manuscript. Page-5, line 110: “X-ray diffraction data associated with each position were collected in transmission mode by varying the Φ angle while keeping the other angles fixed ($2\theta=0^\circ$, $\omega=0^\circ$ and $\chi=54.74^\circ$).” However, the orientation with respect to the conodont element is not relevant. The data themselves demonstrate which is c and which is a. This is basic mineralogy. It makes zero difference how the crystal is arranged within the machine, or within the conodont. The point the reviewer is trying to make is simply not valid here. If we were trying to discuss how they are arranged within the element, then yes, but at this point in the manuscript we are simply talking about the unit cell itself. That is all.

STATE OF KNOWLEDGE AND DISCUSSION

There is quite some misattribution of data in the citations, which is partially due to a conceptual mixup of nanocrystals and crystallites. For example, references 20 and 21 are cited as the source of information about the size of nanocrystals in conodont tissue, whereas they do not describe or even mention nanocrystals at all. The methods used in these studies are not capable of discerning nanocrystals. The studies which employed appropriate methods and yielded this type of data are not mentioned at all (Pietzner et al. 1968; Nemliher & Kallaste 2012; or in articles cited elsewhere in the manuscript but not in this context: Ferretti et al 2017). Taking these studies into account would substantially change the interpretations and the formulation of the study question.

Thanks for pointing out about that missed citation of those important works. We have cited works by Pietzner et al. 1968 and Nemliher & Kallaste 2012 in appropriate section according to your suggestion. Work by Ferretti et al 2017 is less relevant to the current topic and was on X-ray diffraction of neo-crystals that grows during diagenesis on element surface, but we have cited their work in relevant section. However, we respectfully disagree that these studies substantially change interpretation of particular research questions that’s been addressed in our present work.

Lines 209-210: “albid crown tissue is made of large compact crystals arranged along the c-axes parallel or sub-parallel to the long axis of the element” - the long axis of a coniform element is not the same as the long axis of e.g. pectiniform element, therefore this citation cannot be automatically transposed onto current study; considering the geometry of the elements, it would be c-axes perpendicular to the long axis of the studied taxon.

We appreciate your careful review. The sentence has been rewritten into “Previous studies on simple coniform elements suggested that albid crown tissue is made of large compact crystals arranged along the c-axes perpendicular or sub-perpendicular to the long axis of the element, such that the entire crown can be treated as single homogeneous prism of crystallites [20,22].”

MATERIAL

*The study attempts to link measured parameters with conodont functional morphology, but the chosen conodont is nearly the worst possible taxon for the task, as it has no apparatus reconstruction in the sense of positional homology and element types as referred to in the manuscript (Sa, Sb-c, M) do not express homology and therefore cannot be linked to functional models in better understood conodonts even by analogy. The only functional model available for coniform conodonts, Murdock et al. 2013 10.1098/rspb.2013.1524, is not mentioned, although it would provide criteria for the interpretation. The mechanics of coniform conodonts, and of *Dapsilodus* of all of them, are the least understood of all euconodonts. Given that the method is not destructive, it is confusing that the authors did not address a taxon with an apparatus reconstruction and an occlusal model, or at least one that wouldn't cause the issues with tissue differentiation, e.g. a younger taxon with reduced proportions of albid matter.*

We make no attempt whatsoever to discuss functional morphology. Nowhere do we discuss what each element type was used for, nor how it was interacting with its environment, nor within the feeding apparatus of the animal. Furthermore, we respectfully disagree that this is the “worst possible choice” for a taxon for this study. The lack of positional homology knowledge for comparison with other conodonts due to a lack of apparatus reconstruction is irrelevant. At no point are we suggesting that what is true of Sa elements in our study must automatically mean this is true for all Sa elements of all conodonts, and the same is true for all other elements. The point is that we can easily identify variation between element types in the single species studied here. The lack of positional homology knowledge compared to other conodonts does not change this fundamental fact...that different elements are crystallographically different in this species.

The discussion of a need of functional models to “better understood conodonts” is similarly misguided. It does not matter at all if the elements of *D. obliquicostatus* were functional homologous, functionally analogous, or completely different to other conodonts. If we tried to mandatorially extrapolate our findings to all conodonts, then yes this would be an important thing to point out, but that has no bearing on our conclusions. The exact same thing can be said about the reviewer's comments about occlusal models. How the apparatus of *D. obliquicostatus* occluded has absolutely zero impact on our conclusions.

Choosing a taxon with reduced proportions of albid matter would make the identification of the differences between tissue more difficult, not less difficult.

Finally, *D. obliquicostatus* is the absolutely ideal taxon for this study. This work required us to have multiple elements from each element type from a single species be present in a small handful of samples over a short stratigraphic interval AND have multiple ontogenetic stages of each element type as well. Finding anywhere near all of these requirements for any given species is beyond exceptionally difficult. The unique abundance of *D. obliquicostatus*, combined with its simple morphology, provide the ideal test bed for this study.

The criteria for identifying ontogenetic stage wouldn't be reproducible for other researchers: "ontogenetic development was determined by appearance (size, robustness, apparent wear, transparency)": size is very hard to assess for the reader as the elements in Fig 1 are not scaled to the same magnification; apparent wear is known not be reliable given that conodonts periodically grew new tissue throughout their life; transparency is mostly due to peculiarities of preservation and is known to vary from sample to sample; robustness is rather vague altogether). Some quantitative measure of ontogenetic stage (size, if nothing better is available) should be introduced.

We agree that the figures in the earlier version were not sufficient. We have included a new figure (Fig. 1) illustrating growth series, as well as a better version of the conodont comparison figure (Fig. 2) that is presented synoptically. We thank the reviewer for these suggestions. We believe that any researcher looking at this population of conodonts would have come to precisely the same conclusion about ontogeny in these samples.

Sincerely,

Tori Z. Forbes
Associate Professor